# Kinesin-1, -2, and -3 motors use family-specific mechanochemical strategies to effectively compete with dynein during bidirectional transport

Allison M Gicking[1], Tzu-Chen Ma[1], Qingzhou Feng[1], Rui Jiang[1], Somayesadat Badieyan[2], Michael A Cianfrocco[2], William O Hancock[1]*

[1]Department of Biomedical Engineering, Pennsylvania State University, University Park, United States; [2]Department of Biological Chemistry and the Life Sciences Institute, University of Michigan-Ann Arbor, Ann Arbor, United States

**Abstract** Bidirectional cargo transport in neurons requires competing activity of motors from the kinesin-1, -2, and -3 superfamilies against cytoplasmic dynein-1. Previous studies demonstrated that when kinesin-1 attached to dynein-dynactin-BicD2 (DDB) complex, the tethered motors move slowly with a slight plus-end bias, suggesting kinesin-1 overpowers DDB but DDB generates a substantial hindering load. Compared to kinesin-1, motors from the kinesin-2 and -3 families display a higher sensitivity to load in single-molecule assays and are thus predicted to be overpowered by dynein complexes in cargo transport. To test this prediction, we used a DNA scaffold to pair DDB with members of the kinesin-1, -2, and -3 families to recreate bidirectional transport in vitro, and tracked the motor pairs using two-channel TIRF microscopy. Unexpectedly, we find that when both kinesin and dynein are engaged and stepping on the microtubule, kinesin-1, -2, and -3 motors are able to effectively withstand hindering loads generated by DDB. Stochastic stepping simulations reveal that kinesin-2 and -3 motors compensate for their faster detachment rates under load with faster reattachment kinetics. The similar performance between the three kinesin transport families highlights how motor kinetics play critical roles in balancing forces between kinesin and dynein, and emphasizes the importance of motor regulation by cargo adaptors, regulatory proteins, and the microtubule track for tuning the speed and directionality of cargo transport in cells.

*For correspondence:
woh1@psu.edu

Competing interest: The authors declare that no competing interests exist.

## Editor's evaluation

In their important study, Gicking et al., study the physical properties of artificial complexes composed of the dynein-dynactin-BicD2 (DDB) complex linked to one of three classes of kinesins (1, 2, or 3) via a DNA scaffold. They find that all three kinesins can move to the plus-end of microtubules when coupled to the DDB complex. This is surprising because motors in the kinesin-2 and kinesin-3 families have been shown to have a higher load sensitivity; however, the authors show that the faster reattachment kinetics of these motors compensate for their faster detachment rates under load. This work is compelling and relevant to both the biophysics and the neuroscience fields.

## Introduction

Neurons are elongated, highly polarized cells that require robust, long distance, bidirectional cargo transport to function (*Hirokawa and Takemura, 2005*). Better understanding of the molecular mechanisms underlying bidirectional cargo transport is needed, as disrupted transport in neurons is linked

**eLife digest** Nerve cells in the human body can reach up to one meter in length. Different regions of a nerve cell require different materials to perform their roles. The motor proteins kinesins and dynein help to transport the required 'cargo', by moving in opposite directions along tracks called microtubules. However, many cargos have both motors attached, resulting in a tug-of-war to determine which direction and how fast the cargo will travel. In many neurodegenerative diseases, including Alzheimer's, this cargo transport goes awry, so a better understanding of exactly how this process works may help to develop new therapies.

There are three families of kinesin motors, for a total of about a dozen different kinesins that engage in this process. Motors in each of the three families have different mechanical properties. Specific cargos also tend to have specific kinesins attached to them. Here Gicking et al. hypothesized that when pulling against dynein in a tug-of-war, kinesins from the three families would behave differently.

To test this hypothesis, Gicking et al. linked one kinesin to one dynein motor, one at a time in a test tube, and then observed how these two-motor complexes moved using fluorescence microscopy techniques. Unexpectedly, kinesins from the three different families competed similarly against dynein: there were no clear winners and losers. By incorporating previously published data describing the different motor behaviors, Gicking et al. developed a computational model that provided deeper insight into how this mechanical tug-of-war works.

The modeling indicated that kinesins from the three families use different approaches for competing against dynein. Kinesin-1 motors tended to pull steadily against dynein, only detaching relatively rarely, but then take some time to attach back to the microtubule track. In contrast, kinesin-3 motors detached easily when they pull against dynein, but they attach back to the microtubule track quickly, taking only about a millisecond to start moving again. Kinesin-2 motors exhibited an intermediate behavior.

Overall, these experiments suggest that the mechanical properties of the motor proteins are not the main factors determining the direction and speed of the cargo. In other words, the outcome of this molecular tug-of-war does not necessarily depend on which motor is stronger or faster. Rather, further mechanisms, including regulation of the adapter molecules that connect the motors to their cargo, may help to regulate which cargo go where in branched nerve cells. A better knowledge of how all these different factors work together will be important for understanding how cargo transport in nerve cells is disrupted in neurodegenerative diseases.

to neurogenerative diseases including Alzheimer's, hereditary spastic paraplegia, and amyotrophic lateral sclerosis (*Bilsland et al., 2010*; *Chevalier-Larsen and Holzbaur, 2006*; *De Vos et al., 2008*; *Gabrych et al., 2019*; *Millecamps and Julien, 2013*; *Stokin and Goldstein, 2006*; *Ström et al., 2008*). Intracellular cargo is carried by the cytoskeletal motors kinesin and dynein, which move in opposite directions, toward the plus-end and minus-end of microtubules, respectively (*Vale, 2003*). Interestingly, it has been shown that kinesin and dynein are simultaneously present on axonal vesicles (*Encalada et al., 2011*; *Hendricks et al., 2010*; *Maday et al., 2012*; *Sims and Xie, 2009*; *Soppina et al., 2009*), suggesting that successful bidirectional transport depends on coordination between, and strict regulation of, these antagonistic motors.

The predominant model for bidirectional transport is the 'tug-of-war' model, which posits that if both kinesin and dynein are present, they will pull against each other, and the strongest motor will determine the cargo directionality (*Gross, 2004*). Notable experimental support is the elongation of endosomes immediately preceding a directional switch in *Dictyostelium* cells (*Soppina et al., 2009*). However, results from a body of experimental and computational studies suggest that the tug-of-war model is not sufficient to account for the range of bidirectional transport activities in cells. Multiple studies have observed a 'paradox of codependence' (*Hancock, 2014*), wherein inhibiting plus-end-directed motors abolishes minus-end-directed movement instead of enhancing it, and vice versa (*Gross et al., 2002*; *Kunwar et al., 2011*; *Martin et al., 1999*). This complexity suggests mechanisms beyond pure mechanical tug-of-war, such as a requirement of both motors for full activation, cargo binding, and regulation.

Cellular studies of bidirectional transport have provided important information about the specific isotypes and numbers of motors present on cargo during transport (*Cason et al., 2021*; *Hendricks et al., 2010*; *Shubeita et al., 2008*), as well as characterizing cargo dynamics (*Barkus et al., 2008*; *Kamal et al., 2000*; *Levi et al., 2006*; *Maday et al., 2012*; *Rosa-Ferreira and Munro, 2011*; *Tanaka et al., 1998*), and the codependence of opposite directionality motors (*Gross et al., 2002*; *Kunwar et al., 2011*; *Martin et al., 1999*). However, these studies are limited in their ability to decouple inherent motor properties from external regulation via cargo adaptors, microtubule associated proteins (MAPs), and other factors. In vitro optical trap studies have provided precise measurements of the force generation capabilities of single as well as teams of motors (*Andreasson et al., 2015b*; *Budaitis et al., 2021*; *Gennerich et al., 2007*; *Guydosh and Block, 2006*; *Hendricks et al., 2012*; *Rai et al., 2016*; *Sanghavi et al., 2021*). However, in recent work, traditional single-bead optical trap experiments have been shown to impose non-negligible vertical forces on motors that may accelerate their detachment rate under load (*Khataee and Howard, 2019*; *Pyrpassopoulos et al., 2020*). One remedy for this problem is the three-bead trap assay, used widely in studies of myosin (*Finer et al., 1994*), which significantly minimizes vertical forces and provides a more physiologically relevant measurement of motor behavior under load (*Howard and Hancock, 2020*; *Pyrpassopoulos et al., 2020*). Still, in these three-bead traps, the movement of a gliding microtubule is being measured and direct tracking of the motor is difficult. Therefore, there is a need for assays that precisely measure motor behavior under physiologically relevant loads and without extra confounding variables.

To directly track kinesin and dynein motor pairs in vitro, an elegant method was developed that fuses single-stranded DNA to each motor, links them together through complementary DNA base pairing, and tracks the motor pairs by two-color total internal reflection fluorescence (TIRF) microscopy (*Belyy et al., 2016*). This method has been used extensively to investigate the mechanical competition between kinesin-1 and various activated dynein complexes bound to BicD2 (DDB), BicDR1 (DDR), and Hook3 (DDH) (*Belyy et al., 2016*; *Elshenawy et al., 2019*; *Feng et al., 2020*; *Ferro et al., 2020*). These studies report that while DDB can substantially slow down the stepping of kinesin-1, kinesin-1 still dominates kinesin-DDB transport. On the other hand, DDR and DDH, which are more likely to contain two dyneins and may more effectively activate dynein (*Grotjahn et al., 2018*; *Urnavicius et al., 2018*), pull kinesin-1 toward the minus-end more often than DDB (*Elshenawy et al., 2019*). In neurons and other cells, dynein complexes also transport cargo against members of the kinesin-2 (*Hendricks et al., 2012*; *Hendricks et al., 2010*; *Loubéry et al., 2008*) and kinesin-3 (*Schuster et al., 2011*) families. Importantly, kinesin-2 and -3 families have motility and force generation properties that are distinct from kinesin-1 (*Andreasson et al., 2015b*; *Arpağ et al., 2019*; *Budaitis et al., 2021*; *Chen et al., 2015*; *Feng et al., 2018*; *Lessard et al., 2019*; *Mickolajczyk and Hancock, 2017*; *Shastry and Hancock, 2010*; *Zaniewski et al., 2020*). Kinesin-1 can withstand substantial hindering forces for long durations, meaning it is not prone to detaching under load (*Blehm et al., 2013*; *Pyrpassopoulos et al., 2020*; *Schnitzer et al., 2000*; *Visscher et al., 1999*). In contrast, members of the kinesin-2 and -3 families have been shown to rapidly detach under load (*Andreasson et al., 2015b*; *Arpağ et al., 2019*; *Budaitis et al., 2021*). Interestingly, these kinesin-2 and -3 motors have also been shown to reengage with the microtubule and resume force generation at faster rates than kinesin-1, perhaps compensating for the rapid detachment (*Andreasson et al., 2015b*; *Arpağ et al., 2019*; *Budaitis et al., 2021*; *Feng et al., 2018*). Despite these detailed mechanistic descriptions, it remains unclear how the different motile properties of these diverse kinesins affect their coordination with dynein complexes during bidirectional transport and what role these diverse motility properties play in fast axonal transport.

A recent computational study that used a stochastic stepping model to simulate bidirectional transport found that the different properties of kinesin-1 and -2 motors substantially affected the directionality and velocity of cargo transport with DDB (*Ohashi et al., 2019*). There were three key results from this study: (i) the magnitude of the stall force for either the kinesin or DDB motors had only a small effect on the overall cargo velocity, (ii) DDB-kinesin-1 pairs had an average cargo velocity near zero while DDB-kinesin-2 pairs had an average cargo velocity of −300 nm/s, and (iii) the sensitivity of detachment to load was the strongest determinant of the net cargo velocity. It is notable that despite DDB having a lower stall force parameter than kinesin-2 in these simulations (4 pN versus 8 pN), DDB-kinesin-2 pairs had primarily DDB-directed cargo motility (average velocity of −300 nm/s) in the simulations. Overall, these results suggest that force-dependent detachment, rather than stall force,

is the best metric for predicting bidirectional transport behavior of a particular set of motor pairs or teams, but this hypothesis needs to be experimentally confirmed.

In the current study, we reconstituted DDB-kinesin bidirectional transport by linking one kinesin and one DDB motor together with complementary single-stranded DNA, and directly tested how the different motility properties of members of the kinesin-1, -2, and -3 families impacted the resulting bidirectional motility of DDB-kinesin complexes. Surprisingly, we found that, when analyzing events where both motors were engaged and moving on the microtubule, the kinesin-2 and -3 motors were able to withstand hindering loads from DDB nearly as well as kinesin-1. A stochastic stepping simulation of the three motor pairs support a mechanism by which the fast reattachment kinetics of kinesin-2 and -3 counteracts their rapid detachment under load and enables robust force generation against DDB motors. These results confirm the idea that load-dependent detachment and reattachment are the key parameters that determine motor performance under load and point to family-specific mechanochemical strategies to achieve successful cargo transport.

## Results

### Reconstituting DDB-kinesin bidirectional transport in vitro

To create DDB-kinesin motor pairs, we first expressed and purified constitutively active C-terminal SNAP tag fusion constructs for well-characterized members of the kinesin-1, -2, and -3 families: *Drosophila melanogaster* KHC, *Mus musculus* KIF3A (KIF3A/A), and *Rattus norvegicus* KIF1A (*Figure 1A*; further details in Methods). For the remainder of the text, we will refer to these three motors as Kin1, -2, and -3, respectively (*Figure 1A*). The SNAP tags were substoichiometrically functionalized with a 63 bp DNA oligonucleotide and an Alexa Fluor 647 dye to achieve a population of dual-labeled kinesin motors (*Figure 1A*). The DDB complex consisted of full-length recombinant dynein expressed in Sf9 cells, purified dynactin from cow brain, and truncated, recombinant BicD2 (25–424) with a C-terminal GFP tag. We determined the concentration of the oligo-labeled kinesin monomers via an SDS-PAGE shift assay (*Figure 1—figure supplement 1A, C and E*) and then calculated the concentration of labeled dimers using the percent reduction in the unlabeled (unshifted) band intensity of an SDS-PAGE gel after oligo-labeling (*Figure 1—figure supplement 1B, D and F*). We also confirmed that each labeled motor exhibited velocity and run lengths similar to previously published values for similar constructs (*Figure 1—figure supplements 2 and 3*; *Feng et al., 2018*; *Lessard et al., 2019*; *Mickolajczyk and Hancock, 2017*). The functionalized kinesin was linked to the DDB-GFP complex via a GFP nanobody (GBP) (*Kubala et al., 2010*) functionalized with the complementary 63 bp DNA oligonucleotide (*Figure 1B*). Each of the motor pairs were imaged using two-channel TIRF microscopy to simultaneously track both the kinesin and DDB.

The resulting kymographs included populations of free kinesin motors, free DDB complexes, and colocalized pairs. Microtubule directionality was determined via directionality of the free motors (*Figure 1C*). Each set of DDB-Kin traces contained plus-end-directed events with short durations, along with slower events with long durations and net directionality toward either the plus-end or minus-end. Within a single trace, there was considerable velocity heterogeneity, including fast, slow, and paused segments (*Figure 1D*, *Figure 1—figure supplement 4A-F*). Notably, directional switches, defined as sequential segments that move in opposite directions, were rare, occurring with a frequency of 0.01 $s^{-1}$ for all (*Figure 1—figure supplement 4G-I*). To understand the differences between the dynamics of the DDB-Kin1, DDB-Kin2, and DDB-Kin3 pairs, we next quantified the overall velocity for each trace and examined differences between the trace velocity distributions for the three motor pairs.

### DDB-Kin1 pairs move toward plus-end faster and more often than DDB-Kin2/3 pairs

We first asked how the velocity distributions of the motor pairs compared to velocity distributions of a single kinesin or DDB under zero load. We did this by taking the average velocity of each motor or motor complex over the entire run (trace velocity), where a positive velocity is kinesin dominated motility and a negative velocity is DDB dominated motility. For DDB-Kin1 pairs, the median trace velocity was 290 nm/s compared to 586 nm/s for unloaded kinesin-1; this is a 50% decrease but much faster than has been reported previously (*Figure 2A*; *Belyy et al., 2016*; *Feng et al., 2020*). In contrast, the median trace velocities of Kin2 and -3 were –28 and –6.3 nm/s, respectively, which are

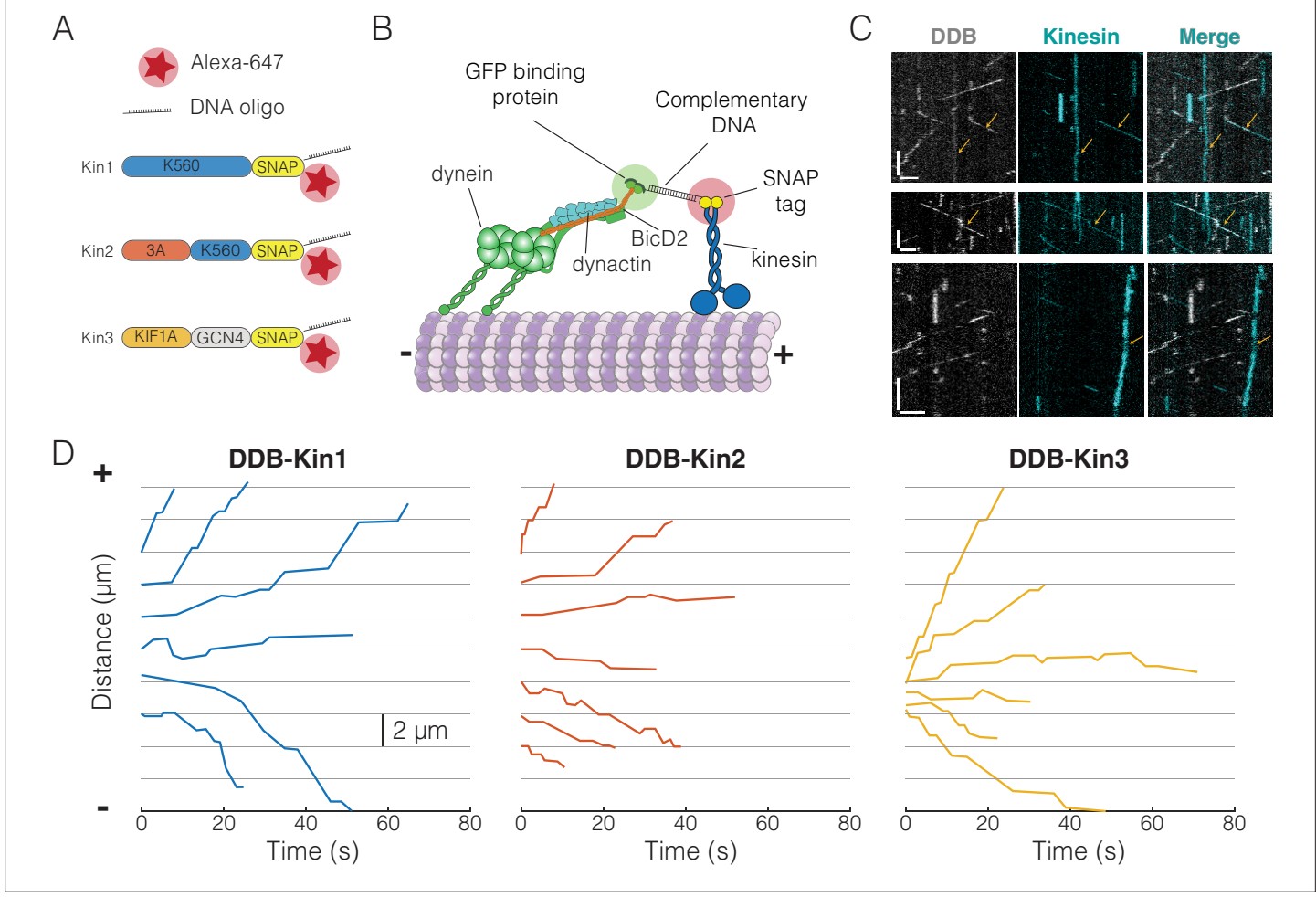

**Figure 1.** Experimental setup and visualization of dynein-dynactin-BicD2 (DDB)-Kin complexes. (**A**) Schematic of kinesin constructs containing SNAP tags functionalized with an Alexa Fluor 647 dye and a single-stranded DNA oligo. (**B**) DDB and kinesin motors connected via complementary DNA oligos on the GFP binding protein (GBP) and SNAP tag. (**C**) Sample kymograph showing the DDB/GFP channel (gray), the kinesin/Alexa Fluor 647 channel (cyan), and the overlay. Scale bars are 2 μm (horizontal) and 10 s (vertical). Microtubule (not shown) is oriented with plus-end to the right. Colocalized events are indicated by an arrow. (**D**) Sample x-t plots for DDB-Kin1, DDB-Kin2, and DDB-Kin3 complexes.

The online version of this article includes the following source data and figure supplement(s) for figure 1:

**Source data 1.** Position vs. time data for the plots in *Figure 1D*.

**Figure supplement 1.** Purification gels and shift assays.

**Figure supplement 1—source data 1.** Purification and shift-assay SDS-PAGE gel images.

**Figure supplement 2.** Unloaded run length and velocity for Kin1/2/3.

**Figure supplement 2—source data 1.** Velocity and run length data for Kin1, Kin2, and Kin3.

**Figure supplement 3.** Unloaded run length and velocity for dynein-dynactin-BicD2 (DDB).

**Figure supplement 3—source data 1.** Velocity and run length data for DDB.

**Figure supplement 4.** Sample traces for dynein-dynactin-BicD2 (DDB)-kin1/2/3 pairs.

**Figure supplement 4—source data 1.** Position vs time data for the sample plots.

dramatically slower than the unloaded motor speeds (*Figure 2A*). Next, we compared the fraction of complexes that moved with net plus- versus net minus-end directionality. We found that 75% of DDB-kin1 pairs had a net plus-end displacement, whereas only 44% of DDB-Kin2 pairs and 49% of DDB-Kin3 pairs had a net plus-end displacement (*Figure 2B*).

Notably, the DDB-Kin1 and DDB-Kin2 trace velocity distributions had two clear peaks, one centered near zero and a second centered near the unloaded kinesin motor velocities (*Figure 2C*).

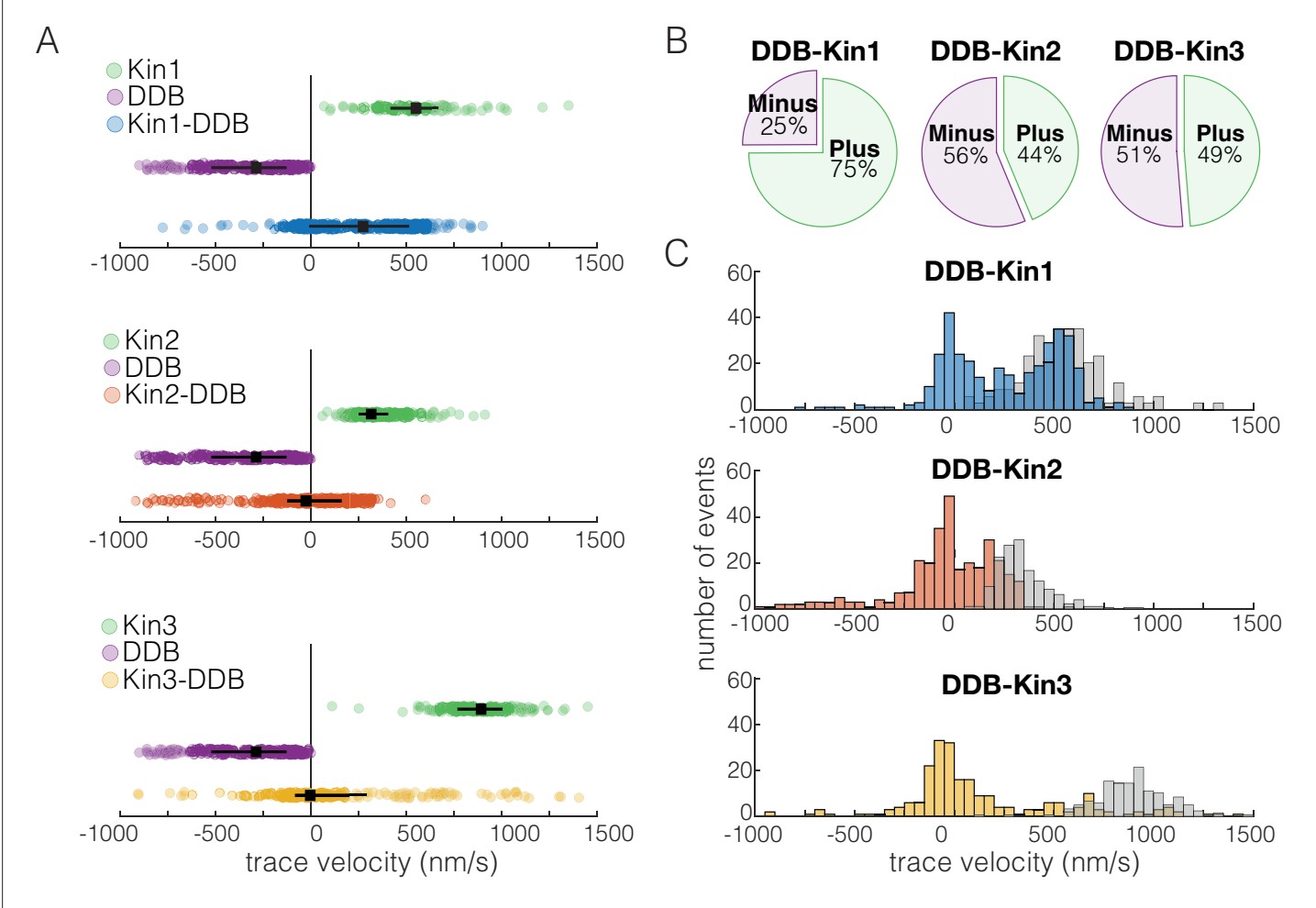

**Figure 2.** Dynein-dynactin-BicD2 (DDB)-Kin1 pairs move faster, and more frequently, to the plus-end than DDB-Kin2/3 pairs. (**A**) Scatter plots showing whole trace velocities of the kinesin alone (gray/top), DDB alone (green/middle), and the DDB-Kin1/2/3 pair (blue/orange/yellow/bottom). Error bars represent median values and quartiles. (**B**) Fraction of motor pairs having net plus-end displacement (dark blue/orange/yellow) or net minus-end displacement (light blue/orange/yellow). (**C**) Histogram of the motor pair velocities. Unloaded kinesin velocity distributions are shown in gray (second histogram on the right). Dashed line indicates fast-slow trace velocity threshold.

The online version of this article includes the following source data for figure 2:

**Source data 1.** Velocity data for DDB-Kin1, DDB-Kin2, and DDB-Kin3.

The DDB-Kin3 velocities had a similar peak near zero, but the fast plus-end population was dispersed rather than centered around a clear peak; this may in part be due simply to the larger range of possible speeds for the faster Kin3. Due to the substantial overlap of the fast plus-end peaks with the velocity distributions for isolated unloaded kinesins, we next investigated whether these two modes represent two configurations of the motor pair: the slow mode representing traces where both kinesin and DDB are engaged on the microtubule and the fast mode representing traces where only the kinesin is engaged.

### Two populations represent only kinesin engaged or both kinesin and DDB engaged on the microtubule

To separate out the fast plus-end population for our analysis, we defined a trace velocity threshold of 250 nm/s for the DDB-Kin1/3 pairs and 125 nm/s for the DDB-Kin2 pairs. We picked these values based on the clear separation of peaks in the trace velocity histograms, and because >95% of the unloaded velocity data lie above this threshold (*Figure 2C*). The fraction of plus-end events above these velocity thresholds was 52% for DDB-Kin1, 29% for DDB-Kin2, and 25% for DDB-Kin3, suggesting that Kin1

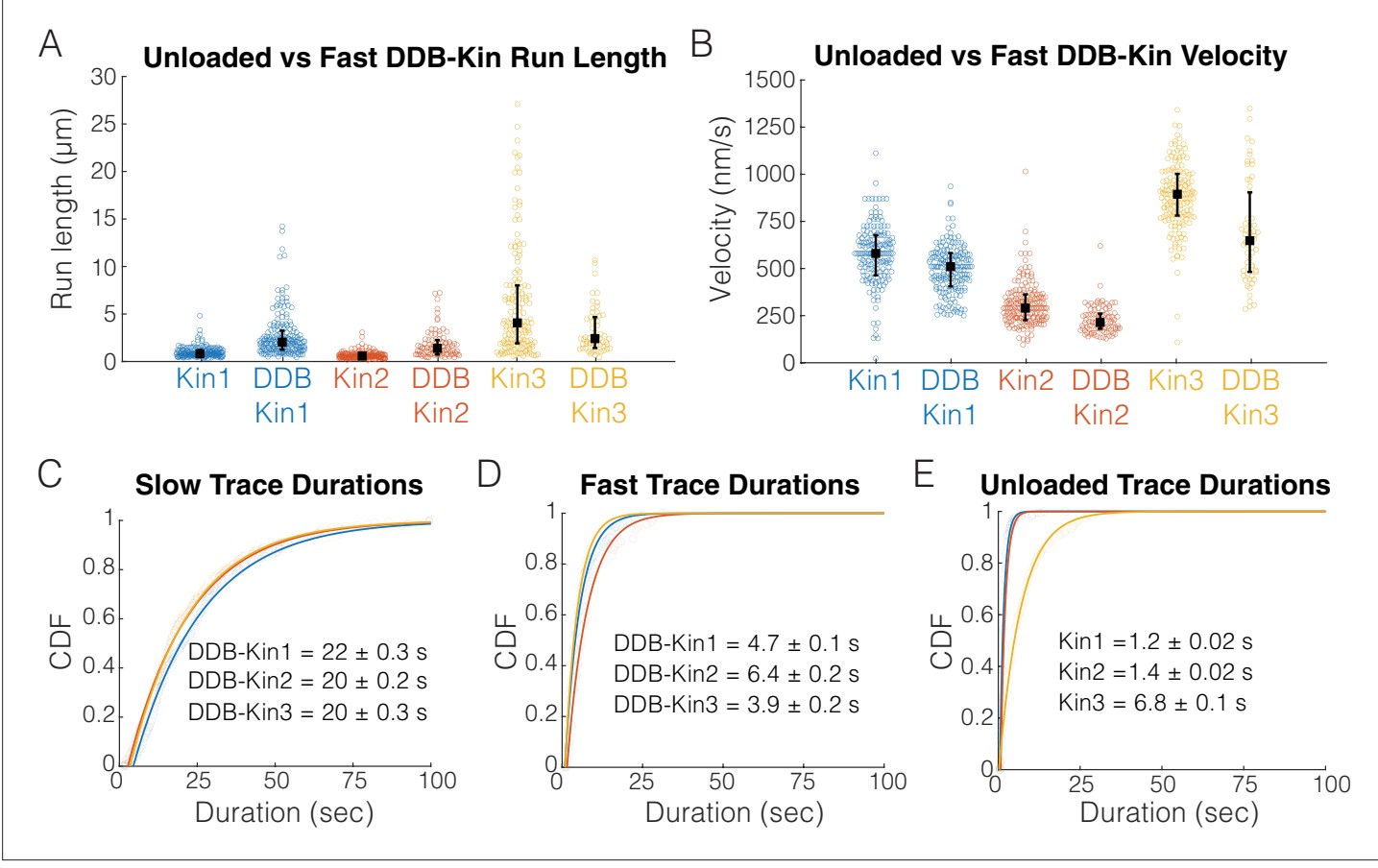

**Figure 3.** Fast, plus-end events represent a diffusive or weakly bound dynein-dynactin-BicD2 (DDB). (**A**) Run length distributions of the fast traces and unloaded kinesin. Error bars represent median values and quartiles. (**B**) Velocity distributions of the fast DDB-Kin traces and unloaded kinesin. Error bars represent median values and quartiles. (**C**) Cumulative distribution functions (CDF) of the durations of the slow traces (traces < velocity threshold). Data were fit to a single exponential and values are mean duration ± 95% CI of bootstrap distributions. (**D**) CDF of the durations of the fast traces (traces > velocity threshold). Data were fit to a single exponential and values are mean duration ± 95% CI of bootstrap distributions. (**E**) CDF of the durations of the unloaded kinesin traces. Data were fit to a single exponential and values are mean duration ± 95% CI of bootstrap distributions.

The online version of this article includes the following source data for figure 3:

**Source data 1.** Run length, velocity, and trace duration data for Kin1, Kin2, Kin3, and the DDB-Kin pairs.

is more likely to pull DDB off the microtubule and move at an unloaded speed than Kin2 or Kin3. However, it is unclear if DDB is fully detached, or remains tethered to the microtubule in a diffusive or weakly bound state.

To determine whether DDB is detached or in a weakly bound state, we next compared the motility of the fast motor pairs (pairs with trace velocities above the thresholds) with the motility of unloaded kinesin in single-molecule assays. Comparing run lengths, we found that for Kin1 and -2, the presence of DDB caused an ~180% enhancement of the run length (**Figure 3A**). This run length enhancement was not observed for Kin3 (**Figure 3A**); however, any potential enhancement is likely masked because very long microtubules were selected for Kin3 alone to achieve a best estimate for the run length (see the long tail for Kin3 in **Figure 3A**), whereas, to maximize the number of events captured in the first few minutes of imaging, DDB-Kin3 events were collected from microtubules of varying lengths. Motor velocities were also affected: coupling with DDB caused a 14% reduction for Kin1 and an ~25% reduction for both Kin2 and -3 (**Figure 3B**). The fact that the velocities were still relatively fast, but DDB had differential impacts on the different kinesin families, is consistent with DDB being in a diffusive or weakly bound state that both tether the kinesin to the microtubule to enhance the run length and create a frictional drag that has a greater effect on Kin2 and -3 motors.

To better understand the differences between the slow and fast DDB-Kin populations, we next compared the trace durations. The mean durations of the slow traces were 22.2±0.3 s for DDB-Kin1, 20.3±0.2 s for DDB-Kin2, and 19.5±0.3 s DDB-Kin3 (mean ± 95% CI of bootstrap distributions of a single exponential fit; *Figure 3C*). These durations are substantially longer than either the durations of the fast DDB-kin populations (*Figure 3D*) or the durations of the unloaded motors (*Figure 3E*), further supporting the idea that the fast population of DDB-Kin traces represent only the kinesin walking on the microtubule with the DDB being in a diffusive or other weakly bound state. It follows that the slow velocity, long duration population of DDB-Kin traces represent cases where both motors are engaged in a strongly bound state. Therefore, to explore more deeply how kinesin and dynein connected to a

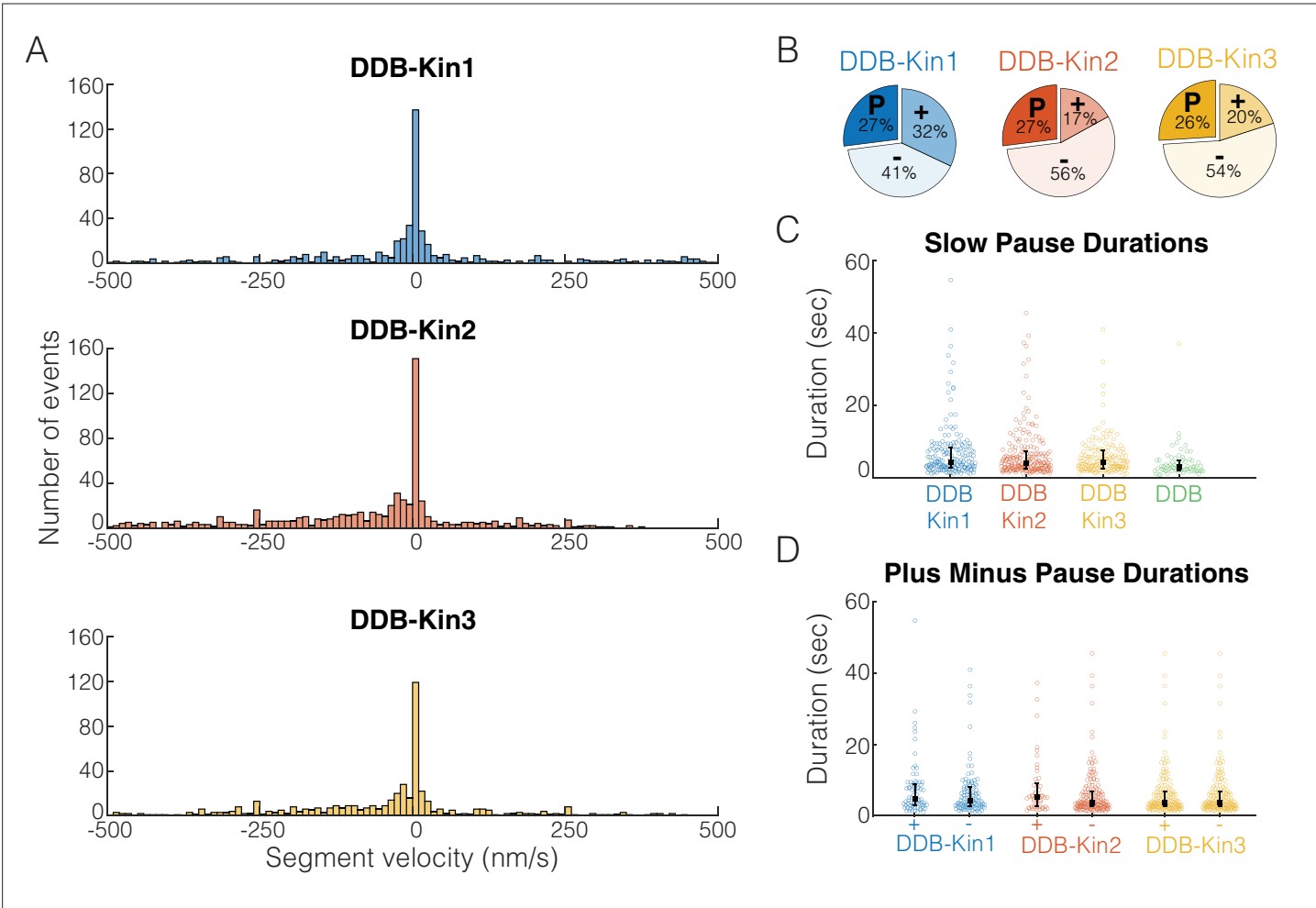

**Figure 4.** Pauses are due to a DDB 'stuck' state.
(**A**) Distributions of segment velocities for the slow traces. Less than 10% of the data is excluded to zoom in on the peak at zero. (**B**) Fraction of segments that are paused (defined as moving <1 pixel/73 nm), minus-end-directed and plus-end-directed for the slow population (<velocity threshold defined in *Figure 2C*). (**C**) Distributions of pause durations for DDB-Kin1/2/3 pairs compared with unloaded DDB. Error bars represent median values and quartiles. (**D**) Comparison of pause durations for the minus-end and plus-end-directed events for each motor pair. Error bars represent median values and quartiles.

The online version of this article includes the following source data and figure supplement(s) for figure 4:

**Source data 1.** Segment velocity and duration data for Kin1, Kin2, Kin3, and the DDB-Kin pairs.

**Figure supplement 1.** Slow segment velocity distributions.

**Figure supplement 1—source data 1.** Cumulative distribution of the segment velocity data for the DDB-Kin pairs.

**Figure supplement 2.** Sample traces for dynein-dynactin-BicD2 (DDB) alone.

**Figure supplement 2—source data 1.** Position vs. time data for DDB alone.

shared cargo compete during bidirectional transport, we focused on the motility of these slow DDB-Kin pairs.

## Pauses observed in motor pairs are due to DDB switching into a 'stuck' state

To understand the behavior of the complex when both motors are engaged, we first segmented each trace into segments of constant velocity and plotted the resulting segment velocity distributions (*Figure 4A*). Interestingly, there were no clear peaks corresponding to the unloaded DDB or kinesin velocities (*Figure 4—figure supplement 1*), suggesting the fraction of time where one motor is detached, or not providing a hindering load, is minimal. However, for all three kinesins, the segment velocity distributions had a clear peak at zero velocity, suggesting that one, or both, of the motors spend a significant fraction of time in a static paused state. If these paused states are caused by periods where the kinesin and DDB are pulling with equal force, we would expect the time spent in these paused states to vary between the three kinesin families due to their different propensities to backstep or detach from the microtubule under load (*Andreasson et al., 2015b*; *Budaitis et al., 2021*; *Feng et al., 2018*; *Ohashi et al., 2019*).

To test whether the fraction of time spent in a paused state was different between the pairs, we quantified the fraction of time that each motor pair spent in each motility state – moving toward the plus-end, moving toward the minus-end, and paused. A paused segment was defined as any segment that moved less than one pixel (73 nm) in either direction, and plus- and minus-end moving segments involved displacements of more than one pixel. We found that, although the fraction of time spent moving toward the plus-end or minus-end varied across the motor pairs, each motor pair spent a similar 26–27% of the time in a paused state (*Figure 4B*). Thus, the fraction of time spent paused is independent of the kinesin type involved and is likely an inherent property of DDB. To confirm the pauses are inherent to the DDB motility, we measured the fraction of time that isolated DDB spends in a paused state and found that it was 24% – almost identical to the motor pairs (*Figure 4—figure supplement 2*). And in further support of this idea, a previous high-resolution tracking study that rigorously characterized DDB state switching reported that unloaded DDB spends 31% of its time on a microtubule in a 'stuck' state (*Feng et al., 2020*).

To further test whether pulling forces by linked kinesin motors affect the DDB paused state, we next asked whether the duration of the DDB pauses was altered when paired with a kinesin. First, we compared the pause durations of the motor pairs with the pause durations of unloaded DDB and found that compared to the unloaded DDB pause segment durations of 2.8±0.08 s (mean duration ± 95% CI of bootstrap distributions), the paused segment durations for the motor pairs were 5.1±0.06 s for DDB-Kin1, 4.5±0.06 s for DDB-Kin2, and 4.3±0.05 s for DDB-Kin3 (*Figure 4C*). The longer durations indicate that the linked kinesin does not pull DDB out of a paused state, and the ~30–45% enhancement in pause duration suggests that pulling forces from linked kinesins may actually stabilize the DDB paused state somewhat. Further support for kinesin forces elongating the DDB pause state was the finding that pauses that interrupted plus-end events were up to 22% longer than pauses that interrupted minus-end-directed events (*Figure 4D*). Based on this slight enhancement in the pause durations between DDB alone and the DDB-Kin pairs, and between plus-end and minus-end-directed events, we conclude the pauses are due to a stabilized version of the DDB 'stuck' state rather than a brief stalemate in the tug-of-war.

## Kin1, -2, and -3 can all effectively withstand DDB hindering loads

Since pause durations suggest that the pauses are due to DDB switching into an inactive static state rather than a property of the mechanical tug-of-war, we decided to focus only on the slow, non-paused segments where both motors are necessarily engaged and stepping along the microtubule. To do this, we separated out the paused segments, and broke the remaining velocity segments into 1 s intervals. The distributions of these 1 s instantaneous velocity intervals were plotted for the DDB-Kin1, DDB-Kin2, and DDB-Kin3 pairs (*Figure 5*) to determine if there were any significant differences between the median speed, spread of the distributions, or the fraction of time that complexes move toward the plus-end.

From this instantaneous velocity analysis, Kin1, -2, and -3 motors are all able to withstand hindering loads generated by DDB, but some subtle differences emerge that suggest different underlying

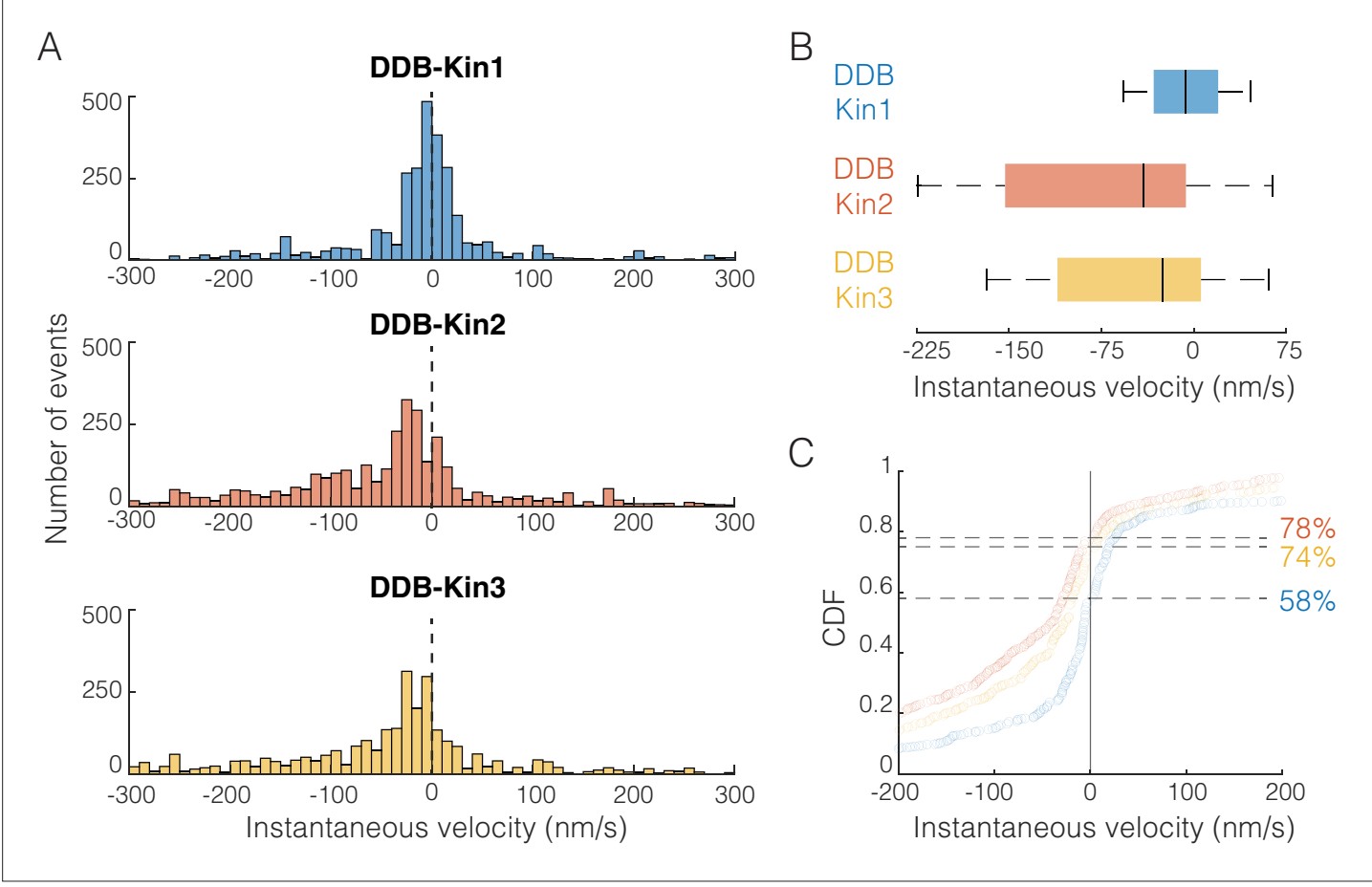

**Figure 5.** Dynein-dynactin-BicD2 (DDB)-Kin1/2/3 all compete effectively against DDB. (**A**) Distribution of instantaneous velocities calculated over 1 s time windows for the moving segments (excluding pauses). Dashed line represents v=0 nm/s. Less than 13% of data are not shown to zoom in on the peak near zero. (**B**) Box plot distributions of the instantaneous velocity distributions shown in (**A**). Vertical bars represent median values (–6, –41, and –26 nm/s), solid boxes represent quartiles, and error bars denote limit of outliers. (**C**) Cumulative distributions of instantaneous velocities, showing the fraction of time spent moving toward the minus-end (<0 nm/s; denoted by dashed lines) versus the plus-end (>0 nm/s). DDB-Kin1 (blue/bottom), DDB-Kin2 (orange/top), and DDB-Kin3 (yellow/middle).

The online version of this article includes the following source data for figure 5:

**Source data 1.** Instantaneous velocity distributions for the DDB-Kin pairs.

mechanisms. First, the peak instantaneous velocity was centered around zero for DDB-Kin1, whereas it was shifted toward the minus-end direction for DDB-Kin2 and DDB-Kin3 (*Figure 5A*). Second, the median speed of DDB-Kin1 was –6 nm/s, which is slower than DDB-Kin2 at –41 nm/s and DDB-Kin3 at –26 nm/s (*Figure 5B*). Third, the DDB-Kin1 velocity distribution was more confined around zero with the 25% and 75% quartiles spanning 52 nm/s, compared to 159 and 117 nm/s for DDB-Kin2 and DDB-Kin3, respectively (*Figure 5B*). Lastly, the fraction of time spent moving toward the plus-end was ~20% higher for the DDB-Kin1 pairs (*Figure 5C*). Together, these data suggest that while all three kinesin motors effectively compete with DDB in tug-of-war, kinesin-1 has a slight advantage. Based on this result, we next wanted to understand how the mechanochemical differences between the three kinesin families lead to their surprisingly similar performances against DDB during bidirectional cargo transport. To do this, we performed stochastic simulations of motor stepping in DDB-Kin pairs.

## Simulations show fast detachment under load can be rescued with fast reattachment

To better understand how the family-specific kinesin motor properties determine the tug-of-war outcomes, we simulated the DDB-kinesin motor pairs using a previously developed stochastic

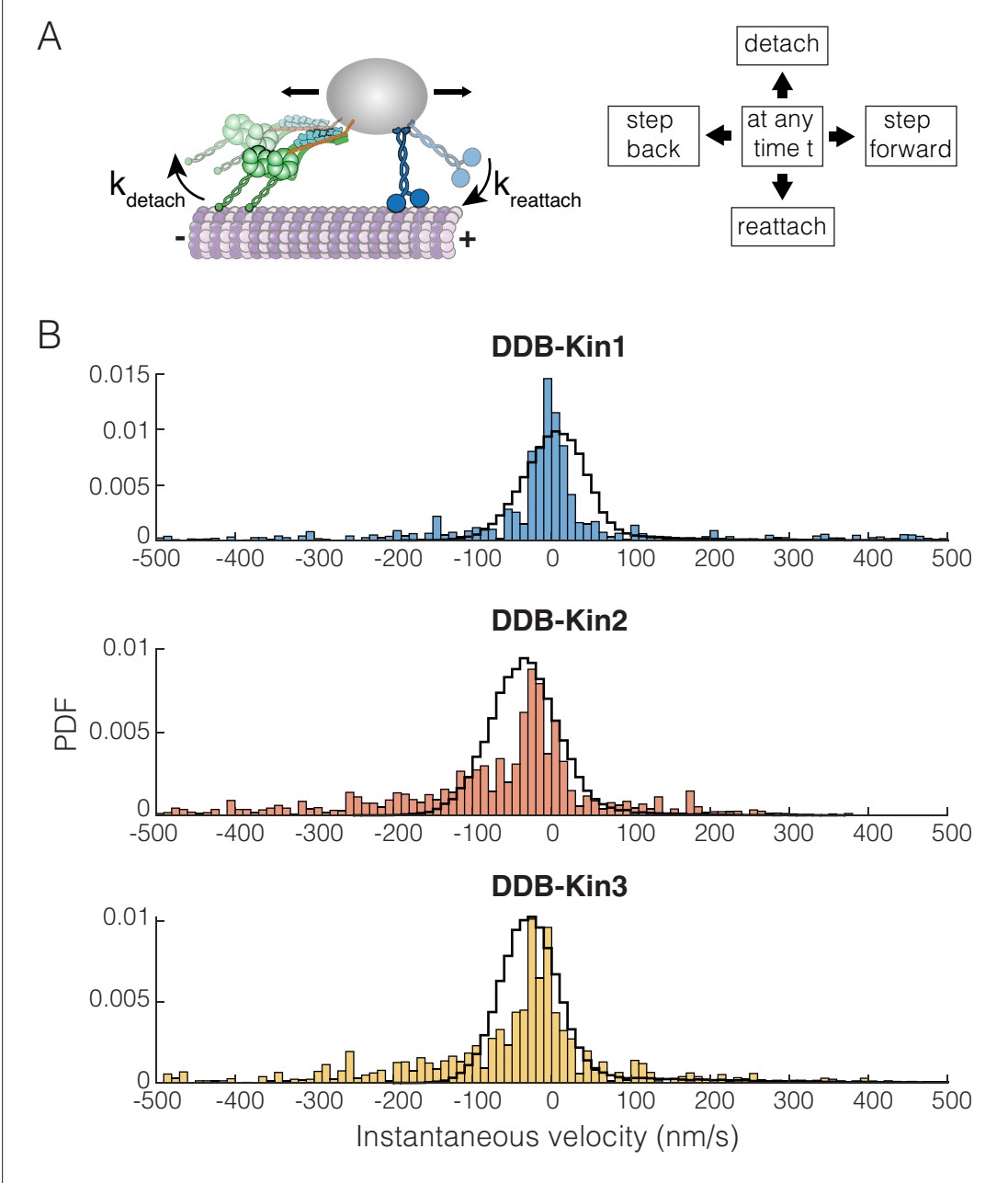

**Figure 6.** Dynein-dynactin-BicD2 (DDB)-Kinesin stepping simulations can recapitulate experimental velocities. (**A**) Schematic of the stochastic stepping model used in simulations. (**B**) Probability density functions (PDF) of the instantaneous velocity distributions of the experimental (blue, orange, and yellow bars) and the simulated traces (black lines) for the DDB-Kin1, DDB-Kin2, and DDB-Kin3 pairs. Window size is 1 s.

The online version of this article includes the following source data, source code, and figure supplement(s) for figure 6:

**Source code 1.** DDB-Kin bidirectional transport simulation.

**Source data 1.** Simulated instantaneous velocity data.

**Figure supplement 1.** Simulation results for slower reattachment rates.

**Figure supplement 1—source data 1.** Simulated position vs. time and instantaneous velocity data.

stepping model of bidirectional transport (*Ohashi et al., 2019*). The model incorporates experimentally determined parameters for kinesin and DDB into a mathematical model that dictates whether each motor will step forward, step backward, detach, or reattach from the microtubule at a given time point (*Figure 6A*; see Methods) (*Ohashi et al., 2019*). The kinesin stepping rates and unloaded

**Table 1.** Model parameters.

| | Kin1 | Kin2 | Kin3 | DDB |
|---|---|---|---|---|
| $V_0$ (nm/s) | 586 | 307 | 910 | 360 |
| $k_{forward}$ (s$^{-1}$) | 76 | 41 | 117 | 60 |
| $k_{detach}^0$ (s$^{-1}$) | 0.96 | 0.76 | 0.16 | 0.1 |
| $F_s$ (pN) | 6 | 6 | 6 | 3.6 |
| $F_{detach}$ (pN) | 6.8 | 3 | 1.3 | $\infty$ (Ideal bond) |
| $k_{reattach}$ (s$^{-1}$) | 100 | 300 | 990 | 5 |
| $k_{backstep}$ (s$^{-1}$) | 3 | 3 | 3 | 5 |
| $\kappa_{motor}$ (pN/nm) | 0.2 | 0.2 | 0.2 | 0.2 |

detachment rates, $k_{detach}^0$, were taken from the single-molecule data in *Figure 3* and *Figure 1—figure supplement 2* (summarized in *Table 1*). The load-dependent motor detachment rate was defined as $k_{detach}(F) = k_{detach}^0 e^{\frac{F}{F_{detach}}}$, where the force parameter, $F_{detach}$, was taken from published optical tweezer experiments (see Methods) (*Andreasson et al., 2015a*; *Andreasson et al., 2015b*; *Elshenawy et al., 2019*). A number of studies have found that kinesin-3 detaches readily under load (*Arpağ et al., 2019*; *Arpağ et al., 2014*; *Budaitis et al., 2021*), and the load-dependent detachment rate for kinesin-3 was taken from a recent study using a three-bead optical trapping assay that minimizes the influence of vertical forces on detachment (*Pyrpassopoulos et al., 2022*). That same study found that kinesin-3 and, to a lesser extent, kinesin-1 frequently disengage under load, slip backward, and then rapidly reengage with the microtubule. This rapid reengagement behavior has also been observed in other recent kinesin-1 studies (*Sudhakar et al., 2021*; *Toleikis et al., 2020*). Based on this work, we used reattachment rates of 100 and 990 s$^{-1}$ for kinesin-1 and -3, respectively. Additionally, based on stopped-flow studies that found the microtubule on-rate constant for kinesin-2 is intermediate between that of kinesin-1 and -3 (*Feng et al., 2018*; *Zaniewski et al., 2020*), we used a reattachment rate of 300 s$^{-1}$ for kinesin-2. These fast rates are further supported by the lack of evidence of long periods of kinesin detachment in any previous studies that tracked DDB-Kin pairs (*Belyy et al., 2016*; *Elshenawy et al., 2019*; *Feng et al., 2020*).

We used the model to simulate bidirectional transport for the three motor pairs and analyzed the resulting trajectories using a 1 s window to calculate instantaneous velocities. Following these initial simulations, we made small adjustments to the parameters to optimize the fits; see *Table 1* for final model parameters. Importantly, we found that the model was able to closely match the experimental instantaneous velocity distributions for each case (*Figure 6B* and *Figure 6—figure supplement 1A*). For DDB-Kin1, the peak simulation velocity of 5 nm/s closely matched the experimental peak of −2 nm/s (*Figure 6B*). For DDB-Kin2 and DDB-Kin3, the velocity peaks were shifted toward the minus-end compared to DDB-Kin1 and the simulations recapitulated this shift but overshot by ~15–20 nm/s toward the minus-end. One aspect of the experimental data that wasn't captured by the models was the minus-end tails in the velocity distributions. These tails are likely due to instances of unloaded DDB movement where the kinesin is detached and is slow to rebind because of an unfavorable conformation or geometry, a feature not incorporated into the model.

To examine the role played by motor reattachment kinetics, we repeated the simulations using a reattachment rate of 5 s$^{-1}$ for all three kinesins. This value was initially determined in a study that measured motor-driven deformations of giant unilamellar vesicles (*Leduc et al., 2004*), it was in a number of modeling studies (*Müller et al., 2008*), and it was experimentally confirmed in a study that used DNA to connect two kinesins (*Feng et al., 2018*). When this 5 s$^{-1}$ value was used for the reattachment rate, the simulated velocities had broad distributions centered around −50,−150, and −250 nm/s for DDB-Kin1, -2, and -3, respectively, strongly conflicting with the experimental data (*Figure 6—figure supplement 1B*). Overall, the data support the conclusion that Kin2 and -3 have higher sensitivity to load than Kin1, as seen in the detachment force parameter and in the larger minus-end shift for Kin2 and -3 when the slower reattachment rate is used in simulations, but this propensity to detach under load is compensated by fast rebinding to the microtubule. By balancing these attachment and detachment kinetics, all three motors can effectively compete with activated dynein motors during bidirectional transport.

## Discussion

Precise determination of motor behavior under physiologically relevant loads, particularly in the context of bidirectional transport of antagonistic motor pairs, is crucial to understanding how bidirectional

transport is regulated in cells (*Cason and Holzbaur, 2022*; *Hancock, 2014*). In neurons, members of the kinesin-1, -2, and -3 families are present on cargo alongside dynein (*Hendricks et al., 2012*; *Hendricks et al., 2010*; *Loubéry et al., 2008*; *Schuster et al., 2011*), but it is unknown why different kinesin motor types are needed and how their differing mechanochemical properties affect their function. Here, we demonstrated experimentally that Kin1 is only slightly more resistant to detaching under load than Kin2 or Kin3, and that in their constitutively active form, all three kinesin motor types generate sufficient force to effectively compete against a DDB complex. This behavior is seen clearly in *Figure 5* where, although the DDB-Kin2 and DDB-Kin3 peaks are wider and shifted to the left from DDB-Kin1, suggesting more frequent detachment of these kinesin, the median speeds are still close to zero. This similarity is surprising, as previous computational modeling work found that the strongest determinant of directionality in DDB-kinesin bidirectional transport is the sensitivity of motor detachment to load (*Ohashi et al., 2019*), and published studies have established that, whereas kinesin-1 is able to maintain stepping against hindering loads, kinesin-2 and -3 motors detach more readily under load (*Andreasson et al., 2015b*; *Arpağ et al., 2019*; *Arpağ et al., 2014*; *Budaitis et al., 2021*; *Pyrpassopoulos et al., 2022*). Thus, it was expected that kinesin-1 would best compete with DDB, and kinesin-3 would compete the least effectively with DDB.

To understand the unexpected experimental results, we used a stochastic stepping model to simulate the DDB-Kin bidirectional transport. We were able to best reproduce the experimental velocity distributions by implementing fast motor rebinding rates in our stochastic model. Previous work from our lab connected two kinesins using a similar DNA linkage approach to the present work and arrived at a motor reattachment rate of 5 $s^{-1}$ (*Feng et al., 2018*), a value used widely in published modeling studies (e.g. *Müller et al., 2008*). However, as part of that work, we also made a first principles calculation of the predicted kinesin reattachment rate, $k_{reattach} = k_{on}^{Mt} \times [tubulin]$. The bimolecular on-rate for microtubule binding in solution, $k_{on}^{Mt}$, was measured by stopped flow to be 1.1 $\mu M^{-1}$ $s^{-1}$ and the effective $[tubulin]$ was calculated to be 125 $\mu M$ (*Feng et al., 2018*), giving a predicted first-order on-rate of 137 $s^{-1}$. Thus, the fast kinesin reattachment rates of 100–1000 $s^{-1}$ used in the simulations are supported by first principles calculations. The precise values we used in the modeling came from a recent three-bead optical tweezer study that measured the rate that kinesin-1 and -3 motors reengaged and resumed a force ramp following termination by a rapid backward displacement (*Pyrpassopoulos et al., 2022*). The large discrepancy between these measured rates of 5 and ~100 $s^{-1}$ could be due to the presence of assisting (Kin-Kin) versus hindering (DDB-Kin) load, where hindering load may be more likely to optimize fast reattachment due to being pulled back along the microtubule. Importantly, fast reattachment as a strategy to compensate for fast detachment under load explains how seemingly 'force-sensitive' kinesin family members can be robust transport motors.

In pairing different types of kinesin motors with DDB, we also gained important insights into the behavior of activated dynein under load. Previous optical tweezer studies have characterized DDB force generation and its stepping behavior under load (*Belyy et al., 2016*; *Elshenawy et al., 2019*), and unloaded single-molecule assays quantified the switching between processive, diffusive, and static states in unloaded single-molecule assays (*Feng et al., 2020*). However, it remained unclear how the switching dynamics would change under load, or how state switching might affect the overall motility when DDB was paired with an antagonistic motor. When full traces were analyzed (*Figure 2*), there was a sub-population of complexes that moved at the kinesin speeds for each motor pair, which we conclude are most likely due to diffusive or weakly bound DDB complexes. When we performed a segmental analysis, which focuses on pairs with two active and engaged motors (*Figure 4*), we found that DDB state switching kinetics did not change dramatically under load – the DDB-Kin complexes spent ~30% of the time in a static state and the rest of the time in a processive state, similar to DDB alone. The similar duration of the pauses between all three motor pairs suggested that these pauses were due to entirely to DDB switching into a static state, and that this must be a strongly bound state that kinesin cannot pull it out of. Interestingly, the slightly longer duration of these pauses in the DDB-Kin complex suggests that the hindering load provided by the kinesin may actually stabilize this paused state, increasing its duration. Overall, these behaviors suggest that some of the complicated vesicle motility observed in vivo (*Hendricks et al., 2010*; *Rai et al., 2016*) could be due to activated dynein switching between states rather than to a tug-of-war between the kinesin and dynein.

Mechanical tug-of-war, where the direction and speed of bidirectional cargo transport is determined by the stronger team of kinesin or dynein motors (*Gross, 2004*), has been the predominant model for nearly two decades (*Hancock, 2014*). However, consistent with the present work, recent studies that paired a kinesin with an activated dynein through complementary DNA have observed primarily slow and smooth motility, with no obvious periods of motors moving at unloaded speeds and either zero or very few instances of directional switching (*Belyy et al., 2016*; *Elshenawy et al., 2019*; *Feng et al., 2020*). These results are contradictory to what is predicted by the tug-of-war model (*Müller et al., 2008*). Interestingly, similar studies of kinesin-dynein bidirectional transport that link the motor pairs with the cargo adaptors TRAK2 and Hook3, rather than DNA, observe primarily fast and unidirectional motility with either zero or very few instances of directional switching (*Canty et al., 2021*; *Fenton et al., 2021*; *Kendrick et al., 2019*). The lack of frequent directional switching in either context suggests that a tug-of-war is not the primary mechanism for determining directionality. Instead, the contrast between the motility of pairs of constitutively active motors linked via DNA versus full-length motors attached to cargo adaptors suggests that cargo adaptors may regulate transport direction by alternately inhibiting kinesin or dynein from engaging with the microtubule, a mechanism termed 'selective activation' (*Cason and Holzbaur, 2022*). Besides providing a more localized site for regulation, direct attachment of antagonistic motor pairs to a common cargo adapter also provides a natural mechanism to ensure that the numbers of plus-end and minus-end motors attached to a cargo are balanced.

An alternative mechanism by which cargo adapters can regulate cargo directionality is by preventing simultaneous binding of kinesin and dynein to the cargo in the first place. It has been shown that the phosphorylation state (in the case of JIP1; *Fu and Holzbaur, 2013*) or the presence of binding partners (in the case of HAP1; *Twelvetrees et al., 2019*; *Twelvetrees et al., 2010*; *Wong and Holzbaur, 2014*) can cause differential binding of kinesin or dynein to a cargo, a mechanism termed 'selective recruitment' (*Cason and Holzbaur, 2022*). Beyond cargo adaptors, the microtubule track itself can also regulate motor binding via recruitment of MAPs that differentially inhibit and recruit motors across the kinesin superfamily, as well as dynein (*Ferro et al., 2022*; *Monroy et al., 2020*). Additionally, cargo stiffness may also alter the motor detachment/reattachment kinetics and thereby affect the bidirectional transport dynamics. The surprising result from the present work is that these kinesin-1, -2, and -3 constructs can all effectively pull against an activated DDB during bidirectional transport, despite their drastically different motility characteristics under load. This result provides strong evidence that motors are not simply regulating themselves via mechanical competition, and underscores the importance of deciphering the combinatorial MAP/adaptor/motor code that regulates transport in cells.

## Methods
### Plasmid design
The Kin1 construct consists of *D. melanogaster* KHC residues 1–559 (adapted from Addgene #129761) (*Coy et al., 1999*), the Kin2 construct consists of the *M. musculus* KIF3A residues 1–342, followed by the *D. melanogaster* KHC neck coil (345–557) for dimerization (adapted from Addgene #129769) (*Shastry and Hancock, 2010*), and the Kin3 construct consists of the *R. norvegicus* KIF1A residues 1–393, followed by a GCN4 leucine zipper for dimerization (adapted from Addgene #61665; *Norris et al., 2014*). All of the kinesin constructs have a C-terminal SNAP tag followed by a 6x His tag for purification. The GFP binding protein nanobody, GBP, contains an N-terminal SNAP tag and a C-terminal 6x His tag for purification (*Feng et al., 2020*; *Feng et al., 2018*; *Kubala et al., 2010*). The BicD2 consists of an N-terminal 6x His tag, *M. musculus* BicD2 residues 25–424 (*McKenney et al., 2014*), followed by GFP, a SNAP tag, and a Strep Tag II.

The dynein plasmid was prepared as described previously (*Schlager et al., 2014*). In summary, the pACEBac1 expression vector containing the dynein heavy chain (DYNC1H1) fused to His-ZZ-TEV-SNAPf tag (pDyn1) and a plasmid containing DYNC1I2, DYNC1LI2, DYNLT1, DYNLL1, and DYNLRB1 (pDyn2) were recombined with Cre recombinase to generate final donor plasmid. pDyn1 and pDyn2 were generous gifts from Andrew Carter, and all genes were codon-optimized for expression in insect cells.

## Protein expression and purification

The kinesin, GBP and BicD2 constructs were bacterially expressed and grown in 800 ml of Terrific Broth (Sigma-Aldrich, St Louis, MO) at 37°C until the OD = 1–2. Induction was initiated by adding 0.3 mM IPTG and the cultures were left to shake at 21°C overnight. The cells were harvested and spun at 123,000× g to collect the supernatant, which was then purified by nickel gravity column chromatography, as described previously (*Gicking et al., 2019*; *Zaniewski et al., 2020*). Elution buffer contained 20 mM phosphate buffer, 500 mM sodium chloride, 500 mM imidazole,10 µM ATP, and 5 mM DTT. For the BicD2, the final elution peaks were combined, supplemented with 10% glycerol, and flash frozen before storage at –80°C. The concentration was determined using absorbance at 488 nm. The kinesin and GBP constructs were exchanged into ×1 PBS with 1 mM DTT and labeled with DNA and SNAP-Surface Alexa Fluor 647 (NEB, Ipswich, MA) dye directly after elution, as described below. Their concentrations were determined using absorbance at 280 nm.

The dynein baculovirus was prepared from pACEBac1 final donor vector using standard methods. High Five Cells (BTI-TN-5B1-4) insect cells at $2\times10^6$/ml density were infected with passage 2 of virus at 1:100 ratio and harvested 72 hr later. For 10 ml culture, 1 ml of lysis buffer (50 mM HEPES pH 7.4, 100 mM NaCl, 10% glycerol 10%, 0.5 mM EGTA, 1 mM DTT and 0.1 mM ATP, 1 unit Benzonase + SIGMAFAST Protease Inhibitor Tablets + 0.5 mM Pefabloc SC) was used. The cell pellet was lysed using a dounce homogenizer at 25 strikes with a tight plunger, and the lysate was clarified by centrifugation for 88 min at 50k rpm 4°C. Clarified lysate was incubated with 0.5 ml packed beads IgG Sepharose 6 Fast-Flow (GE Healthcare, Chicago, IL) for 2–3 hr at 4°C on a tube roller. After incubation, the beads were collected in a disposable column and washed by 150 ml lysis buffer (50 ml with protease inhibitors and 100 ml without protease inhibitors), followed by a wash with 300 ml DynBac TEV buffer (50 mM Tris-HCl pH 8, 2 mM Mg-acetate, 1 mM EGTA, 250 mM K-acetate, 10% glycerol, 1 mM DTT, 0.1 mM Mg-ATP). The beads were transferred to a 2 ml tube, and 1.5 ml DynBac TEV buffer + TEV protease at final 100 µg/ml was added. After overnight incubation at 4°C on tube roller, the supernatant was cleared from the beads using low-binding Durapore filter (0.22 µm). Eluent was subjected to size-exclusion chromatography with Superose 6 300/10 equilibrated with GF150 buffer (25 mM HEPES pH 7.4, 150 mM KCl, 0.5 mM EGTA, 1 mM DTT). The fractions containing dynein were collected and concentrated with 100 kDa MWCO Amicon filter. Glycerol, at final 10%, was added to concentrated dynein before flash-freezing.

Dynactin was purified from bovine brain following previously described procedures (*Urnavicius et al., 2015*). Fresh cow brains were purchased from a local source and washed immediately with ice-cold PBS at least ×3. With the help of a razor, any significant portions of white matter, blood vessels, membrane, brainstem, and corpus callosum were trimmed away. The collected tissue was washed again ×3 with PBS before lysing. Two-hundred ml ice-cold lysis buffer (35 mM PIPES pH 7.2, 1 mM MgSO$_4$, 0.1 mM EDTA, 0.2 mM EGTA, 0.2 mM ATP-Mg, 1 mM DTT, protease inhibitors [cOmplete tablets, or homebrew cocktail + 1 mM PMSF]), 200 µl antifoam, and clean brain tissues were added to a cold metal blender. Tissues were lysed using the 'pulse' blender function – 15 s on, 15 s rest, repeat ×4 at 4°C. Lysate was transferred to Oakridge tubes and centrifuged for 1 hr at 15,000× g, 4°C. Supernatants from the last step were transferred to Ti45 centrifuge tubes and spun at 40k rpm for 45 min, 4°C. Final clarified lysate was loaded into a SP-sepharose column, 300 ml bed volume, equilibrated with Buffer A (35 mM PIPES pH 7.2, 1 mM MgSO$_4$, 0.1 mM EDTA, 0.2 mM EGTA, 0.1 mM ATP-Mg, 1 mM DTT). Bound proteins were fractionated using a two-phase salt gradient: 0% to 25% buffer B (35 mM PIPES pH 7.2, 1 mM MgSO$_4$, 0.1 mM EDTA, 0.2 mM EGTA, 0.1 mM ATP-Mg, 1 mM DTT and 1 M KCl) for 900 ml, and then 25% to 100% buffer B for 300 ml. A western blot for p150[Glued] was used to determine the fractions with dynactin. Fractions with dynactin were collected, diluted twice into HB buffer (35 mM PIPES-KOH pH 7.2, 1 mM MgSO$_4$, 0.2 mM EGTA, 0.1 mM EDTA, 1 mM DTT), and loaded into an HB-buffer equilibrated MonoQ 16/10 column. Unbound proteins were washed out with 10 CV of HB buffer, and dynactin was fractionated using three linear gradients: 5–15% buffer C (HB buffer + 1 M KCl) in 1 CV, 15–35% buffer C in 10 CV, and 35–100% buffer C in 1 CV. Fractions with dynactin were collected and concentrated to 200 µl using 100 kDa MWCO Amicon concentrators. Lastly, size-exclusion chromatography of concentrated dynactin (with Superose 6 300/10 column in non-reducing GF150) resulted in a peak of dynactin just after the void volume. At this point the dynactin subunits were distinguishable on 12% SDS-PAGE. Dynactin fractions were concentrated with 100 kDa MWCO Amicon concentrators and glycerol was added to the final 10%.

## Functionalizing DNA oligos

Complementary amine-modified 63 bp DNA oligos (IDT) were used. The sequences were /5AmMC12/ GT CAA TAA TAC GAT AGA GAT GGC AGA AGG GAG AGG AGT AGT GGA GGT AGA GTC AGG GCG AGA T (kinesin oligo) and /5AmMC12/AT CTC GCC CTG ACT CTA CCT CCA CTA CTC CTC TCC CTT CTG CCA TCT CTA TCG TAT TAT TGA C (GBP oligo). These oligo designs were adapted from previous work (*Belyy et al., 2016*) and confirmed to have a low probability of forming secondary structures. To functionalize the oligos with BG for SNAP tag binding, 250 µM of each oligo was incubated with 13.28 mM of BG-GLA-NHS (NEB) in 100 mM sodium borate and 50% v/v DMSO. The reaction was then desalted into ×1 PBS supplemented with 1 mM DTT using a PD MiniTrap column (Cytiva, Marlborough, MA). The BG-labeling was confirmed using a 10% TBE-Urea electrophoresis gel, and the BG-oligo concentration was determined via absorbance at 260 nm.

## Labeling kinesin and GBP with oligos

BG-oligos were incubated with the SNAP-fusion kinesin and GBP constructs at a 1.5:1 ratio for 1 hr on ice. For the kinesin constructs, 50 µM of SNAP-Surface Alexa Fluor 647 (NEB) was added and incubated for another 30 min on ice to saturate the remaining SNAP tag binding sites. A second nickel gravity column chromatography purification was performed to separate the labeled protein from the excess BG-oligos and dye. The elution buffer contained 20 mM phosphate buffer, 500 mM sodium chloride, 500 mM imidazole, 10 µM ATP, and 3–5 mM DTT. The fraction of oligo-labeled monomers was determined by the percent reduction in the unlabeled (unshifted) band intensity on an SDS-PAGE gel, and the concentration of oligo-labeled monomers was determined via an SDS-PAGE shift assay using a gradient of complementary oligo concentrations. The concentration of oligo-labeled dimers was then calculated from the fraction and concentration of oligo-labeled monomers. For the kinesin constructs, the fraction of oligo-labeled monomer was ~50%, which minimizes the fraction of dimers with two oligos on the SNAP tag to <25%.

For the Kin3 construct, the final oligo-labeled dimer concentration was estimated to be between 1.2 and 1.7 µM assuming 20–80% oligo-labeling of the monomer. We used the average of 1.4 µM.

## MT pelleting assay

To prepare the motors for imaging, a microtubule pelleting assay was performed to remove inactive motors and remove free GBP after incubation with the oligo-labeled kinesin motors. Unlabeled microtubules were polymerized for 30 min at 37°C in BRB80 supplemented with 1 mM GTP, 1 mM MgCl$_2$, and 10% v/v DMSO. The polymerized microtubules were then diluted in Pelleting Buffer (BRB80 with 100 µM AMPPNP, 10 µM Taxol, 0.3 mg/ml BSA, and 0.8 mg/ml casein) to a final concentration of 1.5 µM. The oligo-labeled kinesin were incubated with oligo-labeled GBP on ice for ~10 min and then added to the diluted microtubules at a concentration of 150–300 nM. The microtubule-motor mixture was incubated at room temperature (21°C) for 10 min and airfuged at 25 psi for 10 min to pellet the microtubules. The pellet was resuspended in Resuspension Buffer (30 mM HEPES, 50 mM potassium acetate, 2 mM magnesium acetate, 1 mM EGTA, and 10% glycerol, supplemented with 10 µM Taxol, 50 µM ATP, 0.3 mg/ml BSA, 1 mg/ml casein, 0.2 mM glucose, and 0.2 mM βME), incubated at room temperature for another 10 min and airfuged at 25 psi for 10 min. The supernatant with the active Kinesin-GBP motors was collected and used for the TIRF experiments. The final concentration of active motors was determined by measuring the 647 fluorescence of the pelleting supe (Unbound Motor), resuspension supe (Active Motor), and the resuspension pellet (Rigor Motor) and using the following formula:

$$C_f = (C_i) * \frac{\text{Active Motor}}{\text{Active Motor} + \text{Unbound Motor} + \text{Rigor Motor}}$$

## TIRF assays and analysis

Single-molecule tracking was performed on a custom built micromirror TIRF microscope with a dual view for two-channel imaging (*Nong et al., 2021*). All experiments were performed at 21°C. Unlabeled microtubules were polymerized using the same process described above. Flow cells were prepared by first flowing in 2 mg/ml casein, followed by full-length rigor kinesin (*Mickolajczyk et al., 2015*). Next, Taxol-stabilized, unlabeled microtubules were flowed in and incubated for 30 s, unbound

microtubule were washed out, and the process was repeated ×2. DDB complexes were formed by combining dynein, dynactin, and BicD2 at a 1:1.5:1 ratio (dynein:dynactin:BicD2), and incubating on ice for 10 min. Kin-DDB pairs linked by complementary oligonucleotides were formed by diluting Kin-GBP and DDB complexes to 1–2 nM and incubating them together at an equimolar ratio on ice. Kin-DDB complexes were then introduced into the flow cell in the presence of ~5 µM ATP, allowed to incubate for 2 min, and this process was repeated ×1–2. This low ATP approach maximizes the number of Kin-DDB bound to the microtubules, while retaining activity of the motors. To initiate motility, imaging solution was introduced, consisting of 30 mM HEPES, 50 mM potassium acetate, 2 mM magnesium acetate, 1 mM EGTA, and 10% glycerol, supplemented with 2 mg/ml casein, 20 mM glucose, 37 mM βME, glucose oxidase, catalase, 10 µM Taxol, and 2 mM ATP.

Images were taken at 3.5 fps for 100 s using a Teledyne Photometrics (Tucson, AZ) Prime 95B sCMOS camera. The two channels from the Dual View split screen were aligned using TetraSpeck microspheres (Invitrogen, Waltham, MA). The composite kymographs were then analyzed manually using Fiji (*Schindelin et al., 2012*). Consistent directionality of at least three free kinesin and/or DDB motors were required to determine the microtubule polarity. A trace was determined to be a motor pair event if it could be clearly detected in both channels or if it was a single color moving in the wrong direction (e.g. Alexa Fluor 647/kinesin moving toward the minus-end). Whole trace velocities and run lengths were determined by measuring the distance and time over which the moving complex could be observed. All three motor pairs were prepared and imaged on the same day to control for DDB activity, experiments were repeated on three different days to confirm the trend and then the kymographs were pooled together for further analysis. Velocity segmentation of full traces was done manually, where each segment had to be at least three frames (858 ms) to be counted, and the minimum detectable velocity change between segments was ±10 nm/s. Pauses were defined as any segments that moved less than one pixel (73 nm). Directional switches were defined to be sequential segments that moved at least one pixel in opposite directions. Instantaneous velocity distributions were obtained by weighting each velocity or pause segment by its duration (rounded to the nearest second) and plotting the resulting 1 s segments as a histogram. All of the analysis and plotting was done in MATLAB.

## Simulations

The simulations used a Gillespie stochastic stepping model based on an algorithm developed in a previously published model (*Ohashi et al., 2019*). The reactions of each motor in the simulation were forward stepping, backward stepping, detaching from the microtubule, and reattaching after detachment. Motors having opposite directionality were connected through linear springs with identical stiffness $k_{stiff}$ (representing both the distal regions of the motors and the DNA linker) to a zero-mass, zero-volume cargo. The spring stiffness for each motor was set to 0.2 pN, based on a 6 pN force stretching the motor-DNA assembly to 30 nm, the approximate contour length of the kinesin-DNA and DDB-DNA structures. A companion simulation study found that changing the motor stiffness did have an effect on the velocity distribution, and so further investigation into the role of motor-cargo stiffness is warranted (*Ma et al., 2022*). The load, F, applied on each motor was calculated based on extension ($\Delta$l) and stiffness ($k_{stiff}$) of each motor-DNA linkage as follows:

$$F = k_{stiff} * \Delta l$$

The sign of the load applied on each motor was defined by the direction of motor stepping. For plus-end directed kinesin, assisting loads were positive and hindering loads were negative, and for minus-end-directed DDB, assisting loads were negative, and hindering loads were positive. For kinesin, the unloaded forward stepping rate, $k_{forward}^0$, and the dependence of the forward stepping rate on force (F) were calculated from experimental values for unloaded velocity ($V^0$), step size ($L_{step}$), stall force ($F_{stall}$), and backward stepping rate $k_{back}$, as follows:

$$k_{forward}^0 = \frac{V^0}{L_{step}} + k_{back}$$

$$k_{forward}(F) = \left(k_{back} - k_{forward}^0\right) * \frac{F}{F_{stall}} + k_{forward}^0, F \leq 0$$

$$k_{forward}(F) = k_{forward}^0, F > 0$$

The unloaded kinesin velocities in the simulations were taken from the experimental results in *Figure 2A*. For all three kinesin families, the backstepping rates were set to a uniform 3 s⁻¹, the step size was set to 8 nm, and the stall forces were set to an identical 6 pN. The kinesin load-dependent detachment rate, $k_{detach}$ (F) was modeled using the Bell model (*Bell, 1978*), incorporating the experimental unloaded velocity ($V^0$) and run length ($RL^0$) for each motor, as follows:

$$k_{detach}^0 = \frac{V^0}{RL^0}$$

$$k_{detach}\left(F\right) = k_{detach}^0 * e^{\frac{F}{F_{detach}}}$$

Here, $k_{detach}^0$ is the unloaded detachment rate and $F_{detach}$ is the detachment force parameter, which was shown previously to be a major determinant of the bidirectional transport velocity (*Ohashi et al., 2019*). In this formulation, $F_{detach} = k_B T/\delta$ , where δ is the more commonly used distance parameter measured in optical tweezer studies, and $k_B T$ = 4.1 pN-nm. For kinesin-1, $F_{detach}$ was set to 6.8 pN based on δ=0.6 nm from Figure 6 of Andreasson et al. that used the same kinesin-1 construct as in the present study (*Andreasson et al., 2015a*). For kinesin-2, $F_{detach}$ was set to 3 pN based on δ=1.3 nm from Table S4 of *Andreasson et al., 2015b*, which used the same KIF3A-KHC construct used in the present study. For kinesin-3, $F_{detach}$ was set to 1.3 pN based on *Pyrpassopoulos et al., 2022*, which used the same KIF1A-KHC construct used in the present study. In that work, the unloaded velocity of 1.2 μm/s and run length of 6.3 μm correspond to an unloaded off-rate of 0.19 s⁻¹, and the median engaged time under load was 0.069 s, corresponding to an off-rate of 14.5 s⁻¹. Approximating the force at detachment to be 5–6 pN, this yields an $F_{detach}$ of 1.3 pN.

The DDB stepping model followed a linear force-velocity relationship, similar to kinesin. The DDB detachment rate was independent of load, based with optical tweezer results from yeast dynein that show a minimal change in the off-rate over the range of forces generated by kinesin (*Ezber et al., 2020*). In the simulations, when one motor detaches, the cargo position shifts to the position of the remaining attached motor; similarly, detached motors reattach at the current position of the cargo. All simulations were run 1000 times and each run was recorded for 50 s or until both motors detached from microtubule. All the simulation parameters are listed in *Table 1* in the main text. For comparing simulations with experimental data, the cargo position was averaged every 286 ms, corresponding to the camera frame rate, and the instantaneous velocity was calculated using a 1 s window. A deeper exploration of the sensitivity of the results to the kinesin-1 and DDB parameters and a more detailed description of the model can be found in a recent simulation study from our group (*Ma et al., 2022*).

## Acknowledgements

The authors would like to thank Anthony V Ludlam from the Cianfrocco lab for purifying the dynactin.

## Additional information

### Funding

| Funder | Grant reference number | Author |
| --- | --- | --- |
| National Institutes of Health | R35GM139568 | William O Hancock |
| National Institutes of Health | R01GM076476 | William O Hancock |
| National Institutes of Health | R21AI152869 | Michael A Cianfrocco |
| National Institutes of Health | F32GM137487 | Allison M Gicking |
| National Institutes of Health | T32GM108563 | Rui Jiang |

| Funder | Grant reference number | Author |
|---|---|---|

The funders had no role in study design, data collection and interpretation, or the decision to submit the work for publication.

## Author contributions

Allison M Gicking, Data curation, Software, Formal analysis, Funding acquisition, Investigation, Methodology, Writing - original draft, Writing - review and editing; Tzu-Chen Ma, Data curation, Software, Formal analysis, Investigation, Methodology, Writing - review and editing; Qingzhou Feng, Rui Jiang, Resources, Methodology, Writing - review and editing; Somayesadat Badieyan, Resources, Validation; Michael A Cianfrocco, Resources, Supervision, Methodology, Writing - review and editing; William O Hancock, Conceptualization, Supervision, Funding acquisition, Methodology, Project administration, Writing - review and editing

## Author ORCIDs

Allison M Gicking ⃝ http://orcid.org/0000-0002-9287-2580
Tzu-Chen Ma ⃝ http://orcid.org/0000-0001-9896-8439
Rui Jiang ⃝ http://orcid.org/0000-0001-6000-8512
Michael A Cianfrocco ⃝ http://orcid.org/0000-0002-2067-4999
William O Hancock ⃝ http://orcid.org/0000-0001-5547-8755

## Decision letter and Author response

Decision letter https://doi.org/10.7554/eLife.82228.sa1
Author response https://doi.org/10.7554/eLife.82228.sa2

## Additional files

### Supplementary files

• MDAR checklist

### Data availability

Source files are included for all figures and figure supplements. Source code is included for the simulations in Fig. 6.

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
