## [Editor Report]

In their important study, Gicking et al., study the physical properties of artificial complexes composed of the dynein-dynactin-BicD2 (DDB) complex linked to one of three classes of kinesins (1, 2, or 3) via a DNA scaffold. They find that all three kinesins can move to the plus-end of microtubules when coupled to the DDB complex. This is surprising because motors in the kinesin-2 and kinesin-3 families have been shown to have a higher load sensitivity; however, the authors show that the faster reattachment kinetics of these motors compensate for their faster detachment rates under load. This work is compelling and relevant to both the biophysics and the neuroscience fields.

---

## [Decision Letter]

**Decision letter after peer review:**

Thank you for submitting your article "Kinesin-1, -2 and -3 motors use family-specific mechanochemical strategies to effectively compete with dynein during bidirectional transport" for consideration by *eLife*. Your article has been reviewed by 3 peer reviewers, and the evaluation has been overseen by a Reviewing Editor and Anna Akhmanova as the Senior Editor. The following individual involved in review of your submission has agreed to reveal their identity: Dana N Reinemann (Reviewer #2).

Essential revisions:

After discussion, the reviewers agreed that although the study could be improved by additional experiments, the additional experiments proposed are outside of the scope of this study and require too much additional work. Therefore, the authors should revise their study to clarify certain points, especially the parameters they are measuring, as well as further discuss the limitations of the study and place the results within the framework of prior publications in the field. This is especially true for the kinesin-2 results. The authors should be very clear that they use the KIF3A/KIF3A homodimers, since the KIF3A/KIF3B and KIF3A/KIF3C heterodimers have been shown to be different from KIF3A homodimers.

For more details, please go through the points raised by each reviewer and edit the manuscript accordingly.

*Reviewer #1 (Recommendations for the authors):*

Specific points:

1. It would be very interesting to see if the addition of LIS1, (or if the use of the dynein mutant that disfavors the inhibited state) affects the results of the DDB-kinesin pairs via changes in dynein activation. This is probably beyond the scope of the present manuscript but worth investigation. This is also relevant to the fairly large fraction of time that DDB alone spends in a "stuck" state (~25%), which seems quite high.

2. Lines 225-27 The explanation for the run-length reduction of Kin3 being a function of MT length was not very convincing and could be addressed experimentally.

3. It should be mentioned in the Discussion that a caveat of these findings is that constitutively active kinesins were used in this study, which may contribute to the lack of directional switching.

*Reviewer #2 (Recommendations for the authors):*

The authors present an intriguing study regarding bidirectional transport of cargoes by kinesin and dynein. They examine the mechanistic details of the "paradox of codependence" between kinesin and dynein motility by coupling dynein-dynactin-BicD2 (DDB) and kinesins -1, -2, and -3 via a DNA linker and observing motility using TIRF. Kinesins from different families with differing single molecule motility properties were predicted to differentially affect bidirectional transport, but all three kinesins are able to effectively withstand hindering loads from DDB. Experimental results extracted from unloaded motility assays were compared to simulations to understand the kinesin family-specific implications on observed transport properties. The study is rigorous, well-written, and will be of interest to a wide biophysical audience. I would support the publication of this manuscript upon adequate response to the questions/comments below.

1. I would make it clear in the introduction that you are examining pairs of one kinesin type and one DDB. It wasn't clear to me until the methods and seeing the assay schematic in Figure 1 that there was one of each motor type. Although, changing the number of motors on each side would be interesting as well.

2. The authors use a 63 bp DNA linker to connect DDB and the respective kinesins. Have the authors considered changing the length and/or stiffness of the DNA linker to see how the linker physical and mechanical properties will influence the level of motor cooperation exhibited in this study? Example of changing linker stiffness is in Shrivastava et al., Biochemistry, 2019.

3. To add on question 2, a paper recently published (Xie and Wang, Journal of Physical Chemistry Letters, 2022) modeled dynein and kinesin bidirectionality and also concluded that their collective motility cannot be described with a tug-of-war model. They formulated a "cooperation and competition" model to simulate bidirectional transport. To describe the degree of coupling between motors, the force between cargo and motors was modeled as elastic with a corresponding stiffness k. This parameter appears to better help recapitulate experimental bidirectional data. This leads me to believe that the linker properties may influence the level of cooperation between DDB and kinesin and may also parse out kinesin family-specific dependencies. The authors should consider these experiments, if feasible in the review timeframe. If not, discussion should be added to the manuscript surrounding these points.

4. The following are optical trapping assay ideas for the authors to consider. It may not be reasonable to do these within this review period, depending on the lab's current capabilities, but perhaps it is a helpful discussion.

The authors point out the shortcomings of single bead optical trapping assays due to non-negligible vertical forces on motors that may accelerate their detachment rate under load. However, trapping coupled with this bidirectional assay would yield kinetic parameters from stepping that would help clarify the motility model. The Hancock lab recently published a paper (Feng et al., MBoC, 2020) that couples the DNA linked DDB-kinesin to a Qdot via biotin-streptavidin. The Qdot could be substituted for a streptavidin bead for an optical trapping assay. However, a bead is going to be much larger than a Qdot and thus may have the interference referenced above, in addition to artificially stiffening the DNA linker; but, stepping data could be compared to the model to see if those vertical forces are indeed non-negligible (assay also similar to Furuta et al., PNAS 2013 with multiple kinesin-14 motors).

A probably tricky alternative would be to make a DNA construct where you have the variable size of the DDB-kinesin linker, but then have an additional, much longer DNA linker for attaching the bead to the complex. This way, the geometry and size of the bead is farther from the motor interaction site. On the other hand, if there happened to be directional switching, it may be hard to observe subsequent events if there is slack in the longer DNA tether.

Or, the assay could be flipped into a three-bead assay where the biotinylated DNA linker between DDB and kinesin is attached to a streptavidin bead, and the motors step along a suspended microtubule. It's possible that attaching the linker directly to the bead may artificially stiffen the linkage and thus may affect cooperation, but the vertical force effects on detachment rates would be decreased according to Pyrpassopoulos et al., Biophys J. 2020.

5. CDF should be defined in Figure 3.

6. PDF should be defined in Figure 6.

7. Figure 6 – The assay schematic has kinesin and DDB attached to a bead. Is that what is being simulated, or are you still using the DNA linker, or does the attachment mechanism even matter in the simulation? This does not appear to be discussed in the text.

*Reviewer #3 (Recommendations for the authors):*

1. To show the relation between DDB and kinesin-2, the authors need to analyze DDB-KIF3A/KIF3B, in addition to DDB-KIF3A/KIF3A.

2. It needs to be clarified and specified what kinesin-2 means in each sentence. Ohashi et al., use KIF3A/KIF3B parameters for simulation. This study shows results obtained from KIF3A homodimers. As two studies analyze different motors, one cannot directly compare results from these studies. The difference is not surprising and unexpected (line 426 and line 435).

3. It is an interesting observation that the run length of kinesin-1 and kinesin-2 are extended by DDB while that of kinesin-3 is shorten by DDB (Figure 3). What causes this difference? Can computational simulation (Figure 6) explain this phenomenon?

4. In figure 6, the authors extended Ohashi et al. Again, it needs to be clarified which parameters of kinesin-2 are used here, those of KIF3A/KIF3A homodimers or KIF3A/KIF3B heterodimers.

5. Discussion (Line 476-508). This reviewer feels the authors describe the results obtained in this study as artificial and is not important in a physiological context.

6. Kinesin-1, -2 and -3 transport different cargos. The motility of these cargos in the cell has been analyzed in detail. Each organelle moves differently in the cell. Can current results explain some of these differences in cargo movement?

7. Abbreviations, Kin1, Kin2 and Kin3, need to be explained. These abbreviations may be confusing because some kinesins from Fungi and Tetrahymena have been named Kin1 and Kin2.

---

## [Author Response]

Reviewer #1 (Recommendations for the authors):Essential revisions:Specific points:1. It would be very interesting to see if the addition of LIS1, (or if the use of the dynein mutant that disfavors the inhibited state) affects the results of the DDB-kinesin pairs via changes in dynein activation. This is probably beyond the scope of the present manuscript but worth investigation. This is also relevant to the fairly large fraction of time that DDB alone spends in a "stuck" state (~25%), which seems quite high.

We agree it would be interesting to add LIS1 to see how further activation of the DDB affects the results. It is unclear if the high fraction of time DDB spends in a “stuck” state is due to incomplete activation or is an inherent property of the motor, as this hasn’t been quantified thoroughly by other labs. Of note, in Elshenawy et al., 2020, they use a “phi mutant” that increases the overall motile fraction but there are still frequent pauses observed – even in the presence of LIS1.

2. Lines 225-27 The explanation for the run-length reduction of Kin3 being a function of MT length was not very convincing and could be addressed experimentally.

We acknowledge that this is not the most rigorous explanation. However, because the control Kin3 run lengths are so long, it is difficult to determine any increase, and this problem was compounded by our choice to maximize the number of DDB-Kin3 events we analyzed by including them from all of the microtubules. For control Kin3, there were many events, and our goal was to achieve the best estimate of the true unloaded run length for the modeling parameters, so we limited our data collection to long microtubules where run lengths are not limited by the microtubule length. The long runs can be seen in the tail up to 30 µm for the Kin3 control data in Figure 3A. Because this possible run length enhancement of Kin3 by DDB is only a minor point, we decided not to carry out new experiments. Instead, we modified the explanation of the experimental details in the text in the Results (Lines 245 to 248). (Note that line numbers refer to main text file with track changes highlighted.)

3. It should be mentioned in the Discussion that a caveat of these findings is that constitutively active kinesins were used in this study, which may contribute to the lack of directional switching.

In response, we added a point in the Discussion (Line 460) noting that these are constitutively active motors. While we agree with the reviewer that using constitutively active kinesins could influence the frequency of directional switching, studies that use full length kinesin bound to dynein activating adaptors also do not see much directional switching (Canty et al., 2021; Fenton et al., 2021; Kendrick et al., 2019). This suggests to us that the directional switching observed in the cell is more likely a result of external regulation (e.g. phosphorylation state or MAPs). This point can be found in the Discussion (Lines 529-546).

Reviewer #2 (Recommendations for the authors):The authors present an intriguing study regarding bidirectional transport of cargoes by kinesin and dynein. They examine the mechanistic details of the “paradox of codependence” between kinesin and dynein motility by coupling dynein-dynactin-BicD2 (DDB) and kinesins -1, -2, and -3 via a DNA linker and observing motility using TIRF. Kinesins from different families with differing single molecule motility properties were predicted to differentially affect bidirectional transport, but all three kinesins are able to effectively withstand hindering loads from DDB. Experimental results extracted from unloaded motility assays were compared to simulations to understand the kinesin family-specific implications on observed transport properties. The study is rigorous, well-written, and will be of interest to a wide biophysical audience. I would support the publication of this manuscript upon adequate response to the questions/comments below.1. I would make it clear in the introduction that you are examining pairs of one kinesin type and one DDB. It wasn’t clear to me until the methods and seeing the assay schematic in Figure 1 that there was one of each motor type. Although, changing the number of motors on each side would be interesting as well.

We added text in the Introduction (Lines 122-123) to clarify the point that we are connecting one kinesin to one DDB motor.

2. The authors use a 63 bp DNA linker to connect DDB and the respective kinesins. Have the authors considered changing the length and/or stiffness of the DNA linker to see how the linker physical and mechanical properties will influence the level of motor cooperation exhibited in this study? Example of changing linker stiffness is in Shrivastava et al., Biochemistry, 2019.

We agree it is definitely possible that both the length and stiffness of the DNA linker could impact the results and we think this is an interesting question for future experiments. For instance, our modeling work in a related study

(https://doi.org/10.1101/2022.08.09.503394) revealed that for very compliant linkages in a kinesin-dynein pair, detachment of one motor can cause a significant recoil of the cargo toward the remaining bound motor. This recoil results in a directional switch that would be undetectable for stiff motor-cargo linkages. In cells, this implies that the adaptor type, cargo size and cargo composition may all influence transport. We added text in the Discussion (Lines 545-546) that brings up the point that motor-cargo stiffness can play a role bidirectional transport dynamics.

3. To add on question 2, a paper recently published (Xie and Wang, Journal of Physical Chemistry Letters, 2022) modelled dynein and kinesin bidirectionality and also concluded that their collective motility cannot be described with a tug-of-war model. They formulated a “cooperation and competition” model to simulate bidirectional transport. To describe the degree of coupling between motors, the force between cargo and motors was modelled as elastic with a corresponding stiffness k. This parameter appears to better help recapitulate experimental bidirectional data. This leads me to believe that the linker properties may influence the level of cooperation between DDB and kinesin and may also parse out kinesin family-specific dependencies. The authors should consider these experiments, if feasible in the review timeframe. If not, discussion should be added to the manuscript surrounding these points.

We share your interest in the role of mechanical stiffness on the transport dynamics. We added a point highlighting the possible role of stiffness in the Discussion (Lines 545-546) of the revised manuscript.

4. The following are optical trapping assay ideas for the authors to consider. It may not be reasonable to do these within this review period, depending on the lab’s current capabilities, but perhaps it is a helpful discussion.The authors point out the shortcomings of single bead optical trapping assays due to non-negligible vertical forces on motors that may accelerate their detachment rate under load. However, trapping coupled with this bidirectional assay would yield kinetic parameters from stepping that would help clarify the motility model. The Hancock lab recently published a paper (Feng et al., MboC, 2020) that couples the DNA linked DDB-kinesin to a Qdot via biotin-streptavidin. The Qdot could be substituted for a streptavidin bead for an optical trapping assay. However, a bead is going to be much larger than a Qdot and thus may have the interference referenced above, in addition to artificially stiffening the DNA linker; but, stepping data could be compared to the model to see if those vertical forces are indeed non-negligible (assay also similar to Furuta et al., PNAS 2013 with multiple kinesin-14 motors).A probably tricky alternative would be to make a DNA construct where you have the variable size of the DDB-kinesin linker, but then have an additional, much longer DNA linker for attaching the bead to the complex. This way, the geometry and size of the bead is farther from the motor interaction site. On the other hand, if there happened to be directional switching, it may be hard to observe subsequent events if there is slack in the longer DNA tether.Or, the assay could be flipped into a three-bead assay where the biotinylated DNA linker between DDB and kinesin is attached to a streptavidin bead, and the motors step along a suspended microtubule. It's possible that attaching the linker directly to the bead may artificially stiffen the linkage and thus may affect cooperation, but the vertical force effects on detachment rates would be decreased according to Pyrpassopoulos et al., Biophys J. 2020.

We thank the reviewer for the ideas and agree that the suggested experiments would enhance the story. Again, we think these are excellent avenues for future work.

5. CDF should be defined in Figure 3.

A definition of CDF has been added to the Figure 3 caption.

6. PDF should be defined in Figure 6.

A definition of PDF has been added to the Figure 6 caption.

7. Figure 6 – The assay schematic has kinesin and DDB attached to a bead. Is that what is being simulated, or are you still using the DNA linker, or does the attachment mechanism even matter in the simulation? This does not appear to be discussed in the text.

Thank you for pointing this out. The schematic is accurate – each motor is attached to the common cargo through a linear spring, and the spring stiffnesses are identical. Following each step or detachment of a motor, a force balance is calculated to update the cargo position. In the revised manuscript, this was clarified in the expanded Simulations section in the methods.

Reviewer #3 (Recommendations for the authors):1. To show the relation between DDB and kinesin-2, the authors need to analyze DDB-KIF3A/KIF3B, in addition to DDB-KIF3A/KIF3A.

We appreciate the suggestion, but we feel that this experiment is beyond the scope of the current study and further, we do not expect KIF3A/B to yield substantially different results. This expectation is based on results from the Andreasson 2015 *Current Biology* paper (http://dx.doi.org/10.1016/j.cub.2015.03.013), where we explored differences between KIF3A/A, KIF3B/B and KIF3A/B in an effort to account for heterodimer stepping as an emergent property of the individual 3A and 3B heads. Simply put, although there were differences between KIF3A/A, KIF3B/B and KIF3A/B, those differences were substantially smaller than differences between these kinesin-2 constructs and kinesin-1. For instance, Figure 4A from Andreasson that shows the run length as a function of load, a motor property that we showed in Ohashi et al., (2019) played an outsized role in the velocity of kinesin-dynein complexes. As can be seen from the expanded panel on the right, there are differences between the three kinesin-2 constructs, but when compared to kinesin-1 on the left, those differences collapse. In other work in that paper, there were some differences in the shape of the force-velocity curve for the three kinesin-2 constructs, but in our simulation work we have found that this feature does not substantially alter the bidirectional transport behavior. As a final point, the very surprising finding in our current simulations was that despite significant differences between the kinesin-1, -2, and -3 motor properties, their performance when linked to DDB were fairly similar.

2. It needs to be clarified and specified what kinesin-2 means in each sentence. Ohashi et al., use KIF3A/KIF3B parameters for simulation. This study shows results obtained from KIF3A homodimers. As two studies analyze different motors, one cannot directly compare results from these studies. The difference is not surprising and unexpected (line 426 and line 435).

We were careful to more precisely define kinesin-2 in our revision, thank you for pointing that out. This can be found in the Results (Lines 148-150) and in the revised Simulations section in the Methods (Lines 762 to 820). However, we disagree that we cannot directly compare results from the current study and that of Ohashi et al., because they use different motor parameters. The text starting in line 426 (initial submission) reads:

“This result is surprising, as previous computational modeling work found that the strongest determinant of directionality in DDB-kinesin bidirectional transport is the sensitivity of motor detachment to load (Ohashi et al., 2019), and published studies have established that, whereas kinesin-1 is able to maintain stepping against hindering loads, kinesin-2 and kinesin-3 motors detach more readily under load.”

The point we are making here is that the Ohashi modeling work demonstrated the general principle that in these bidirectional simulations the load-sensitivity of motor detachment is the most important determinant of the resulting bidirectional transport behavior. This is why the result is surprising. See also our response to point 4 below.

3. It is an interesting observation that the run length of kinesin-1 and kinesin-2 are extended by DDB while that of kinesin-3 is shorten by DDB (Figure 3). What causes this difference? Can computational simulation (Figure 6) explain this phenomenon?

DDB appearing to shorten the kinesin-3 run length is an interesting result – however we think this is an artifact of the way the DDB-kin run lengths were measured rather than a real effect. In the kinesin-3 control experiments, only very long (> 20 µm) microtubules were selected to minimize the fraction of end runs, but this wasn’t possible in the DDB-kin assays due to the use of unlabeled microtubules and wanting to maximize the number of events per video. Also, see response to Reviewer 1, point #2 above.

4. In figure 6, the authors extended Ohashi et al. Again, it needs to be clarified which parameters of kinesin-2 are used here, those of KIF3A/KIF3A homodimers or KIF3A/KIF3B heterodimers.

Thank you for the suggestion. We have clarified and expanded the Simulations section in the Methods to fully detail our parameter selection. Regarding our kinesin-2 parameters, we use the unloaded velocity and run length from the control experiments in Figure 3A and B of the present study. For the detachment force parameter of 3.0 pN, we fortuitously were able to use data from a previous optical tweezer study (Andreasson et al., *Curr Biol*, 2015) that used the same KIF3A-KHC homodimer construct. Therefore, the unloaded velocity, unloaded off-rate, and load-dependent detachment for kinesin-2 are all well constrained by our experimental data. Three other parameters (stall force, linear force-velocity, and backstepping rate) are identical for kinesin-1, -2, and -3, and hence cover both KIF3A-KHC or the native KIF3A/B. Finally, the kinesin-2 reattachment rate is an estimate based on measured rates for kinesin-1 and -3 from optical tweezer data, together with stopped flow data for the three motor families, where the kinesin-2 experiments employed the same KIF3A-KHC construct used in our current study.

5. Discussion (Line 476-508). This reviewer feels the authors describe the results obtained in this study as artificial and is not important in a physiological context.

We respectfully disagree. There are many potential mechanisms underlying the observed bidirectional transport dynamics of vesicles, and thus it is important to rule out as many specific mechanisms as possible. We find that pairs of motors do not recapitulate the dynamics of organelles, thus putting the focus on other mechanisms by which transport is regulated.

6. Kinesin-1, -2 and -3 transport different cargos. The motility of these cargos in the cell has been analyzed in detail. Each organelle moves differently in the cell. Can current results explain some of these differences in cargo movement?

Because to a first approximation the three motor pairs all behave similarly, our results cannot explain observed differences in the motility of cargo transported by the three different kinesin families. Instead, our results highlight the important roles played by external regulation via the microtubule track, MAPs, cargo adaptors, and other mechanisms.

7. Abbreviations, Kin1, Kin2 and Kin3, need to be explained. These abbreviations may be confusing because some kinesins from Fungi and Tetrahymena have been named Kin1 and Kin2.

We agree that the abbreviations are confusing and so we have added text in the Results (Lines 148-150) to help clarify what they mean.